# Spatiotemporal neural dynamics of object recognition under uncertainty in humans

Yuan-hao Wu[1], Ella Podvalny[1†], Biyu J He[1,2,3,4]*

[1]Neuroscience Institute, New York University Grossman School of Medicine, New York, United States; [2]Department of Neurology, New York University Grossman School of Medicine, New York, United States; [3]Department of Neuroscience & Physiology, New York University Grossman School of Medicine, New York, United States; [4]Department of Radiology, New York University Grossman School of Medicine, New York, United States

**Abstract** While there is a wealth of knowledge about core object recognition—our ability to recognize clear, high-contrast object images—how the brain accomplishes object recognition tasks under increased uncertainty remains poorly understood. We investigated the spatiotemporal neural dynamics underlying object recognition under increased uncertainty by combining MEG and 7 Tesla (7T) fMRI in humans during a threshold-level object recognition task. We observed an early, parallel rise of recognition-related signals across ventral visual and frontoparietal regions that preceded the emergence of category-related information. Recognition-related signals in ventral visual regions were best explained by a two-state representational format whereby brain activity bifurcated for recognized and unrecognized images. By contrast, recognition-related signals in frontoparietal regions exhibited a reduced representational space for recognized images, yet with sharper category information. These results provide a spatiotemporally resolved view of neural activity supporting object recognition under uncertainty, revealing a pattern distinct from that underlying core object recognition.

*For correspondence:
biyu.he@nyulangone.org

Present address: [†]Center for Molecular and Behavioral Neuroscience, Rutgers University, Newark, United States

## Editor's evaluation

This study presents valuable findings on the fine spatiotemporal profiles of object recognition in human brains under noisy and ambiguous conditions. The evidence supporting the conclusion is compelling, with state-of-the-art techniques and model-driven fusion of MEG and 7T fMRI. The work will be of broad interest to cognitive neuroscientists working on consciousness and related fields.

## Introduction

We recognize objects in our environment rapidly and accurately across a variety of challenging viewing conditions, allowing us to successfully navigate everyday life. Consider driving a car, it is of utmost importance to our survival that we correctly differentiate between various traffic signs in low visibility conditions, such as during a rainstorm or on a foggy night.

In visual neuroscience, object recognition processing has commonly been viewed and studied as an automatic feedforward process within the ventral visual stream from the primary visual cortex to the inferior temporal cortex (*Riesenhuber and Poggio, 1999*; *DiCarlo et al., 2012*). A wealth of knowledge from neurophysiology and neuroimaging contributes to our understanding about how neural computations unfolding in this feedforward process support 'core object recognition'—our ability to recognize, identify, and categorize objects presented in clear, high visibility conditions. However, how

the brain carries out object recognition tasks when sensory input is noisy or ambiguous, as often is the case in the natural environment, remains poorly understood.

Recent accumulating evidence points to the intriguing possibility that object recognition under increased uncertainty (e.g. occlusion, clutter, shading, crowding, complexity) is supported by an interplay of feedforward and feedback signals between the ventral visual and higher-order frontoparietal regions. For instance, recent primate neurophysiology studies have shown that the ventrolateral prefrontal cortex actively participates in object recognition processing under challenging circumstances (due to occlusion, image complexity, or fast presentation) by sending feedback signals to ventral visual cortices (*Fyall et al., 2017*; *Kar and DiCarlo, 2021*; *Bellet et al., 2022*).

In humans, early neuroimaging evidence using a visual masking paradigm revealed that the orbitofrontal cortex (OFC) might send top-down feedback signals to inferotemporal (IT) cortex conveying a fast initial analysis of the visual input (*Bar et al., 2006*). Importantly, the extent of higher-order frontoparietal involvement in object recognition has been substantially broadened by later work (*Hegdé and Kersten, 2010*; *Filimon et al., 2013*; *Wang et al., 2013*; *González-García et al., 2018*; *Levinson et al., 2021*; *Mei et al., 2022*). For example, recognition of object images presented at liminal contrasts was associated with changes in the activation/deactivation level and object category information in widespread cortical networks (*Levinson et al., 2021*).

These recent findings suggest that object recognition processing under uncertainty is not confined to the ventral visual areas, but is jointly supported by regions at different cortical hierarchical levels. While these studies revealed which brain regions are involved in object recognition processing, they leave open the questions of how recognition-related signals dynamically evolve in each of these brain regions and how they coordinate with each other. For instance, some brain regions may exhibit transient signals while others may manifest sustained responses. Neither is it known how recognition-related signals propagate across the brain: They may evolve concurrently across multiple brain areas or flow sequentially from one area to another. Finally, different brain regions might have different representational formats related to how success or failure of objects recognition sculpts the neural activity patterns and the information carried by them.

In this study, we addressed these open questions by investigating the spatiotemporal dynamics of neural activity underlying object recognition under uncertainty. We combined MEG and ultra-high-field (7 Tesla) fMRI data recorded during a threshold-level object recognition task, and employed a model-driven MEG-fMRI fusion approach (*Hebart et al., 2018*; *Flounders et al., 2019*) to track neural dynamics associated with different aspects of object recognition processing with high temporal and spatial resolution. Our results provide a spatiotemporally resolved view of neural activity supporting object recognition under uncertainty in the human brain, and reveal a picture of rich and heterogeneous representational dynamics across large-scale brain networks.

## Results

### Paradigm and behavior

To investigate the spatiotemporal dynamics of brain activity supporting object recognition under uncertainty, we combined MEG (*N*=24) and fMRI data (*N*=25) obtained from two separate experiments using the same stimuli set and experimental paradigm (with trial timing tailored to each imaging modality, see *Figure 1A*). The main datasets analyzed herein were previously published in *Podvalny et al., 2019*, and *Levinson et al., 2021*.

In each trial, participants viewed a briefly presented image containing an object presented at a liminal contrast. After a delay of variable length, participants reported the object category and their recognition experience (*Figure 1A*). For the recognition question, participants were instructed to report 'yes' whenever they saw a meaningful object in it, and report 'no' if they saw nothing or low-level features only. In the case of an unrecognized image, participants were instructed to provide a genuine guess of the object category. The stimuli set included five unique exemplars from each of the following categories: human face, animal, house, and man-made objects (*Figure 1B*). For each participant, the contrast of stimuli was titrated to reach an ~50% recognition rate using a staircase procedure before the main experiment. In addition, a phase-scrambled image from each category was also included in the experiments; the scrambled images were not included in the analyses reported below due to comparatively low trial numbers.

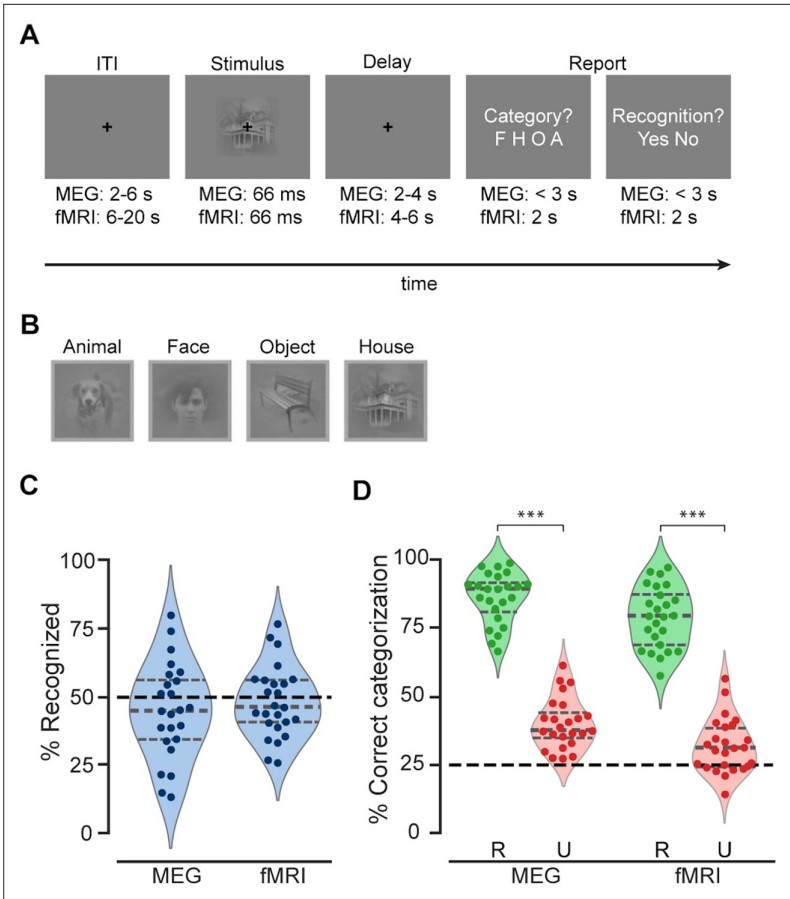

**Figure 1.** Experimental paradigm and behavioral results. (**A**) Trial timeline. (**B**) An example real image from each category. (**C**) Percentage of real-image trials reported as recognized in the MEG and fMRI experiments. Recognition rates were very close to the intended threshold-level recognition rate of 50% in both experiments. (**D**) Categorization accuracy for real images in recognized (R) and unrecognized (U) trials. The black dashed line indicates chance level at 25%. ***: p<0.001; significant differences in categorization behavior between recognized (R) and unrecognized (U) trials (one-sided Wilcoxon signed-rank test). (**C–D**) The horizontal gray lines of the violin plots indicate 25th percentile, median, and 75th percentile, respectively. Dots depict individual participants' behavioral performance in the MEG (N=24) and fMRI experiments (N=25).

The behavioral results from the MEG and fMRI experiments showed consistent patterns: In both experiments, the recognition rates for real images did not differ from the intended rate of 50%: MEG participants reported 44.9±3.6% of the images as recognized (mean ± SEM, $W=105$, $p=0.21$, two-sided Wilcoxon signed-rank test against 50%), and the mean recognition rate in fMRI participants was 48.0±2.6% ($W=126.5$, $p=0.34$, *Figure 1C*). Further, the object categorization behavior for recognized images was highly accurate across both experiments (MEG: 86.1±1.9%, $W=300$, $p=5.96×10^{-8}$; fMRI: 78.8±2.2%, $W=325$, $p=2.9×10^{-8}$; one-sided Wilcoxon signed-rank test against chance level of 25%) and significantly higher than those for unrecognized images (MEG: $W=300$, $p=5.96×10^{-8}$; fMRI: $W=325$, $p=2.9×10^{-8}$; one-sided Wilcoxon signed-rank test; *Figure 1D*). Interestingly, categorization accuracy in unrecognized trials remained above-chance level (MEG: 40.1±1.8%, $W=300$, $p=5.96×10^{-8}$; fMRI: 32±1.9%, $W=251$, $p=0.002$), consistent with previous studies showing above-chance discrimination performance in conditions where the stimulus content did not reach awareness (*Lau and Passingham, 2006*; *Hesselmann et al., 2011*; *Li et al., 2014*).

Together, this pattern of behavioral results suggests that the object images were successfully presented at subjective recognition threshold, and that participants performed the categorization task meaningfully in both recognized and unrecognized trials.

## Sequential emergence of recognition and object category information

We first investigated the timing of recognition state- and object category-related information in the MEG. The presence of both types of information in the current fMRI dataset was established in a previous study, which showed that brain activity in widely distributed cortical networks contains information about the subjective recognition outcome (recognized vs. unrecognized) and category (face/animal/house/man-made objects) of the images (*Levinson et al., 2021*). Here, utilizing the fine temporal resolution of MEG, we assessed *when* such information emerges.

To assess the temporal evolution of brain signals containing information about subjective recognition (recognized vs. unrecognized), we conducted multivariate decoding analyses on event-related fields (DC-35 Hz) at every time point from 500 ms before to 2000 ms after stimulus onset. Specifically, we applied binary support vector machine (SVM) classifiers (*Chang and Lin, 2011*) to discriminate whole-brain sensor-level activity patterns associated with different recognition outcomes. Similarly, to temporally resolve the unfolding of MEG signals containing information about object categories, binary SVM classifiers were applied to decode object category in a pairwise manner (six pairs total) in recognized and unrecognized trials, respectively. For both analyses, decoding accuracy significantly above chance level (50%) indicated the presence of the information of interest in MEG signals.

Recognition outcome could be significantly decoded from 200 to 900 ms after stimulus onset ($p<0.05$, cluster-based permutation test, *Figure 2A*), with peak decoding at 580 ms. Similar to previous fMRI findings (*Levinson et al., 2021*), the decodability of image category was highly dependent on subjective recognition (*Figure 2B*). When images were reported as unrecognized, image category was undecodable throughout the epoch (*Figure 2B*, red). By contrast, when the same identical images were recognized, decoding accuracy for category membership peaked at 290 ms the first time, though it did not survive cluster correction; decoding accuracy reached significance at 470 ms and remained significant until 1060 ms (*Figure 2B*, green). These results reveal long-lasting neural processes elicited by brief stimuli, consistent with earlier studies using threshold-level visual stimuli (*Dehaene and Changeux, 2011*; *Li et al., 2014*; *Baria et al., 2017*). At the same time, they reveal a rise of stimulus-evoked neural processes influencing recognition behavior at ~200 ms, preceding the emergence of category information at ~470 ms.

It is intriguing that category-related information emerged slowly over time compared to earlier studies using high-contrast images (e.g. *Carlson et al., 2013*; *Cichy et al., 2014*). To test whether the delayed responses were due to the low visibility (and increased uncertainty) of the present stimuli, we ran a separate MEG task block wherein participants viewed the same set of images presented at high contrasts (see Methods for details). Consistent with earlier studies, we found that when image visibility was high, there was a fast rise of category information within the first 200 ms after stimulus onset (reaching significance at 110 ms, peaking at 170 ms; *Figure 2—figure supplement 1*). This result confirms that the late onset of category information observed in the main task was due to the more challenging viewing condition, likely reflecting the more time-consuming recurrent processing involving corticocortical feedback activity (*Lamme and Roelfsema, 2000*; *Fyall et al., 2017*; *Kar and DiCarlo, 2021*).

## Dynamic change of representation format over time

The significant recognition outcome decoding emerging at 200 ms indicates that there was a dynamic change in how recognized images were represented relative to unrecognized images. To probe this further, we applied representational similarity analysis (RSA, *Kriegeskorte et al., 2008*) to whole-head MEG signals. At each time point, we computed a representational dissimilarity matrix (RDM) for sensor-level activity patterns across unique images and recognition outcomes. Each RDM had dimensions of 40×40 (5 exemplars×4 object categories×2 recognition outcomes; *Figure 2C*), with each cell populated with a dissimilarity measure (quantified as 1-Pearson's correlation) between activity patterns of an image/condition pair. We applied multidimensional scaling (MDS) to visualize the structure of RDMs by projecting them onto two-dimensional spaces (*Figure 2D*). In the MDS plot, each dot represents an image associated with a particular recognition outcome, and distances between dots reflect the dissimilarities in neural activity patterns.

As expected, there was no discernable dissimilarity structure at stimulus onset: Dissimilarities between most image/condition pairs were close to one (as indicated by the yellow color in the RDM) and dots in the corresponding MDS plot were arbitrarily distributed. By 200 ms after stimulus onset,

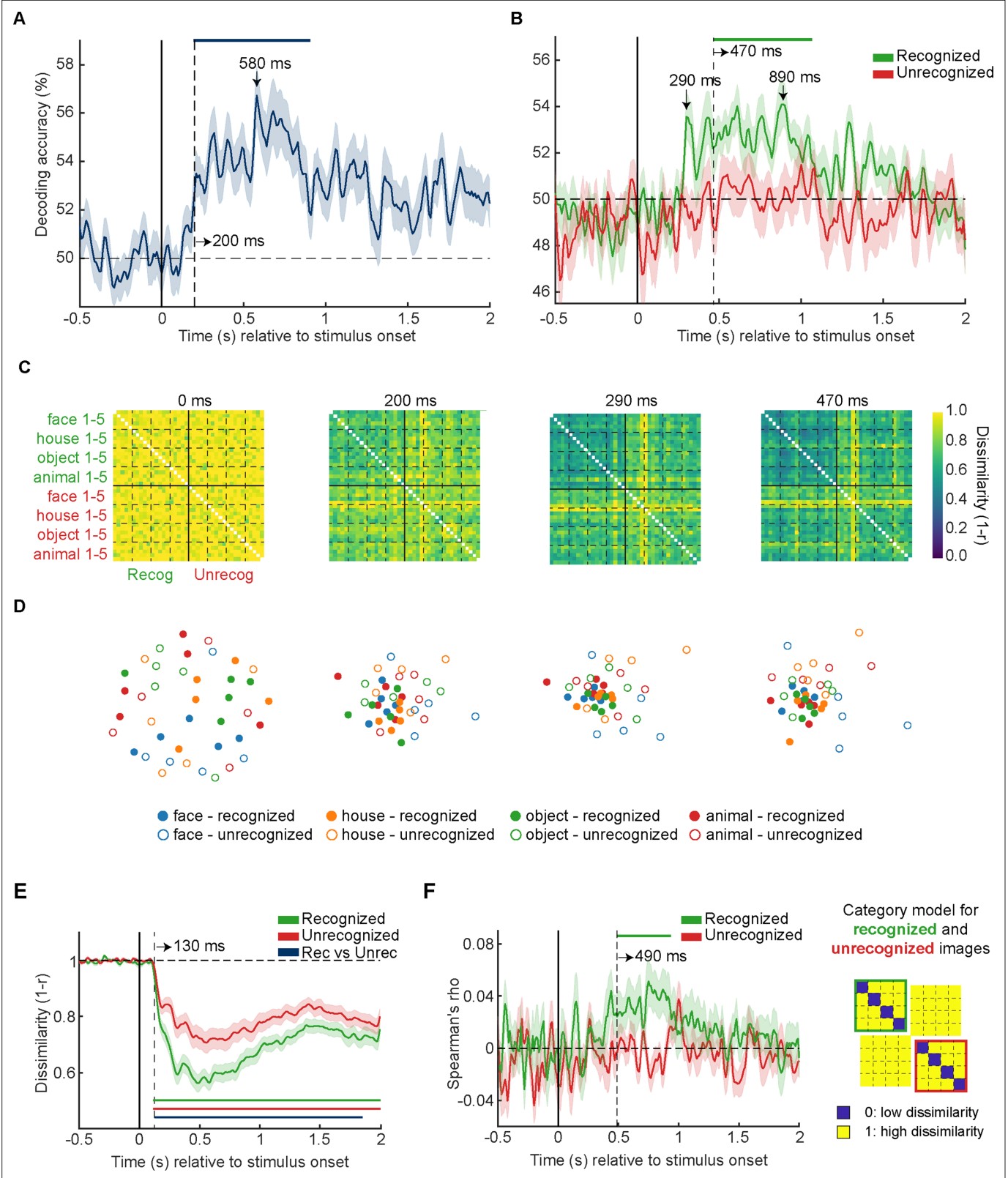

**Figure 2.** MEG multivariate pattern analysis (*N*=24). (**A**) Recognition outcome decoding. (**B**) Image category decoding in recognized (green) and unrecognized trials (red). (**C**) Group-average MEG representational dissimilarity matrices (RDMs) (40×40) at selected time points. (**D**) Visualization of representational dissimilarity structure at the same time points as in (**C**) using the first two dimensions of the multidimensional scaling (MDS). Each of the filled circles represents an exemplar image that was reported as recognized and is color-coded according to its category membership. The

*Figure 2 continued on next page*

*Figure 2 continued*

unfilled points represent the physically identical exemplar images that were not recognized by participants. (**E**) Mean between-image representational dissimilarity in recognized (green) and unrecognized trials (red). (**F**) Left: Correlation between category model RDM and MEG RDMs in recognized (green) and unrecognized (red) trials. Right: Category model RDM. Blue and yellow colors indicates low and high representational dissimilarities, respectively. In (**A–B**) and (**E–F**), shaded areas represent SEM across participants; horizontal dashed lines indicate the chance level, and vertical dashed lines indicate significance onset latencies. Statistical significance is indicated by the colored horizontal bars (p<0.05, one-sided cluster-based sign permutation tests).

The online version of this article includes the following figure supplement(s) for figure 2:

**Figure supplement 1.** Object category decoding in the localizer task.

recognized images had started to cluster together, while unrecognized images remained relatively distributed across the MDS space. To test whether the clustering effect within the recognized images was indeed stronger than unrecognized images, we compared the mean dissimilarity between recognized images against that between unrecognized images at every time point (one-sided Wilcoxon signed-rank tests). As shown in *Figure 2E*, the mean dissimilarity between recognized images (green) was significantly lower than between unrecognized images (red) at most of the post-stimulus time points (130–1900 ms, blue horizontal bar), confirming the prominent clustering effect of recognized images over time.

Together with our report of image category being only decodable in recognized trials but not unrecognized trials (*Figure 2B*), the strong clustering effect of recognized images indicates that neural activity associated with recognized images contained sharper category information compared to unrecognized images despite the overall smaller dissimilarities among each other. To validate the compatibility between category decoding and RDM results, we ran an additional RSA-based analysis to test for the presence of category information in brain responses to recognized and unrecognized images, respectively. At each time point, we compared the representational structure within the recognized images (cells in the upper-left quadrant) and unrecognized images (bottom-right quadrant) with a category model which predicted smaller dissimilarities between images from the same category than between images from different categories (*Figure 2F*, right). This yielded a time course of Spearman's rhos, indexing the strength of category information available in brain responses to recognized and unrecognized images, respectively. As expected, the result was qualitatively similar to category decoding: Significant category information occurred shortly before ~500 ms and decayed at ~1000 ms in recognized trials, whereas no category information was evident in unrecognized trials (*Figure 2F*, left).

Taken together, these findings indicate that successful recognition is associated with a shrinking representational space occupied by cortical activity at 130 ms, which may have a facilitating effect on the formation of object category representation that reaches significance at ~470 ms. In the analyses presented below, we bring in the 7T fMRI data to shed light on the spatiotemporal evolution of this and related neural effects.

## Resolving the spatiotemporal dynamics of recognition-related processes using model-driven MEG-fMRI fusion

The MEG decoding analysis presented above focused on the temporal evolution of recognition-related information at the whole-brain level, leaving open the question of how the related signals unfold in individual brain regions over time. To provide a deeper understanding of the involved neural processes, we applied a model-driven MEG-fMRI fusion approach (*Hebart et al., 2018*; *Flounders et al., 2019*), which provides a spatiotemporally resolved view of different types of neural processes in the brain. This section provides a brief overview of our methodology.

First, we constructed model RDMs that capture two different types of recognition-related processing: (i) Consistent with our MEG results (*Figure 2C–E*) and related prior findings (*Schurger et al., 2015*; *Baria et al., 2017*), the 'recognition model' (*Figure 3A*, left) hypothesizes that successful recognition is associated with a reduction of neural variability, resulting in clustering of activity patterns associated with recognized images (manifesting as low dissimilarity among them). (ii) The 'two-state model' (*Figure 3A*, right), on the contrary, hypothesizes that subjective recognition is associated with a bifurcation in neural signals into two brain states corresponding to recognized and unrecognized

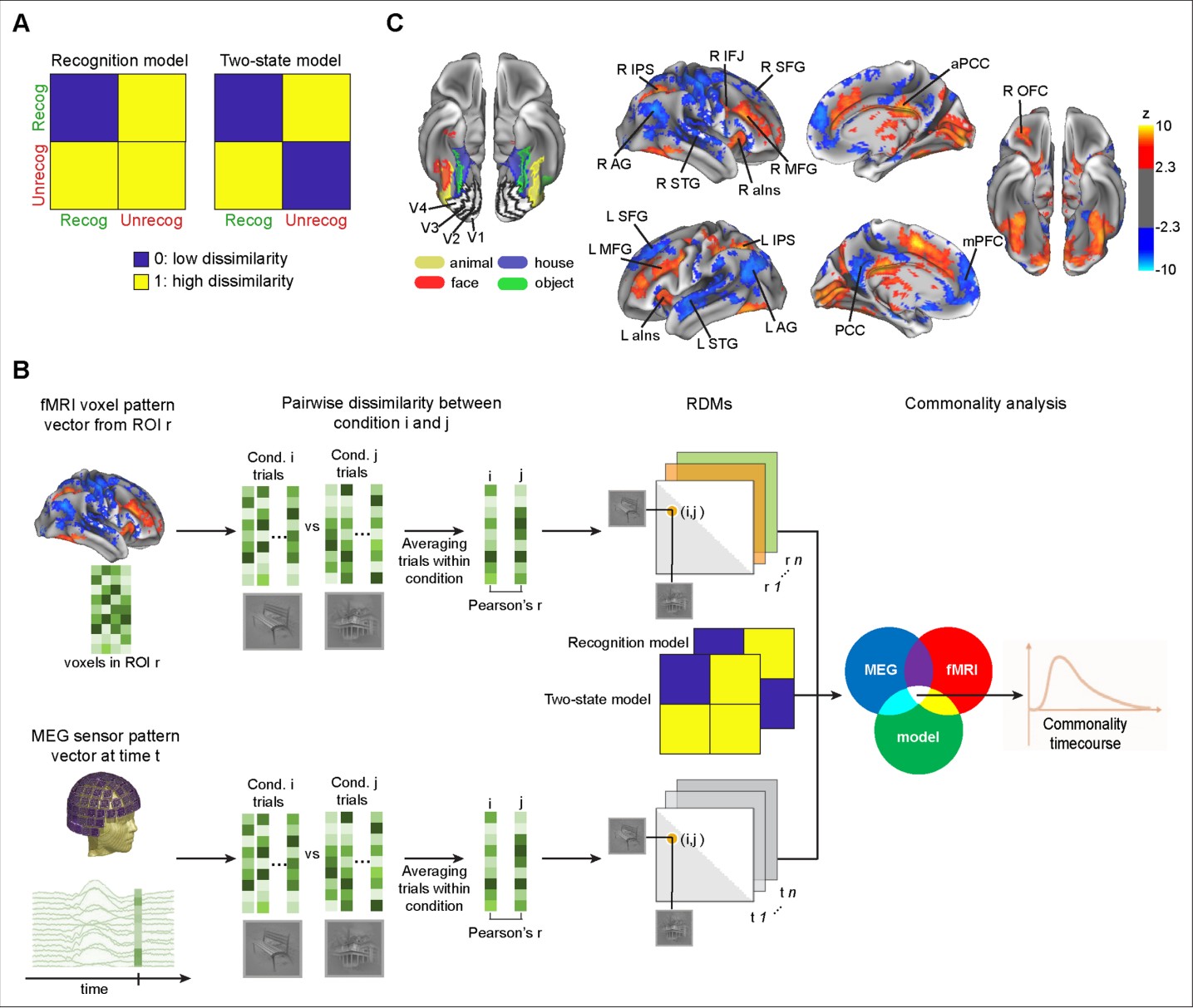

**Figure 3.** Model-based MEG-fMRI fusion procedure. (**A**) Model representational dissimilarity matrices (RDMs) used for model-driven MEG-fMRI fusion. Blue and yellow colors represent low and high dissimilarities, respectively. (**B**) Schematic for the commonality analysis. Commonality coefficient at each time point reflects the shared variance between an MEG RDM of a given time point, fMRI RDM of a given region of interest (ROI), and a model RDM reflecting the expected dissimilarity structure based on a given hypothesis. By repeating the procedure for every single MEG time point, we obtained a commonality time course for a given ROI and a given model RDM. (**C**) fMRI ROIs (from *Levinson et al., 2021*). Left: Early visual cortex (EVC) and ventral temporal cortex (VTC) ROIs from a representative subject, defined using a functional localizer. Right: ROIs outside the visual cortices, in the task-positive network (TPN, orange-yellow colors) and default mode network (DMN, blue), were yielded from the recognized vs. unrecognized general linear model (GLM) contrast. AG: angular gyrus, aIns: anterior insula, aPCC: anterior posterior-cingulate cortex, IFJ: inferior frontal junction, IPS: intraparietal sulcus, MFG: middle frontal gyrus, mPFC: medial prefrontal cortex, OFC: orbitofrontal cortex, PCC: posterior cingulate cortex, SFG: superior frontal gyrus, STG: superior temporal gyrus.

The online version of this article includes the following figure supplement(s) for figure 3:

**Figure supplement 1.** Correlation between MEG and fMRI representational dissimilarity matrices (RDMs) derived from early visual cortex (EVC) and ventral temporal cortex (VTC) regions of interest (ROIs).

**Figure supplement 2.** Same as *Figure 3—figure supplement 1*, with task-positive network (TPN) regions of interest (ROIs).

**Figure supplement 3.** Same as *Figure 3—figure supplement 1*, with default mode network (DMN) regions of interest (ROIs).

outcomes, respectively (manifesting as low dissimilarity between trials with the same recognition outcome and high dissimilarity between trials with different recognition outcomes).

We then used RSA to combine time-varying MEG data with spatially localized fMRI data based on the correspondence between their representational structures, under the assumption that a correspondence between neural measurements by different recording modalities reflects the same neural generators (*Kriegeskorte et al., 2008*; *Cichy and Oliva, 2020*). Specifically, we estimated how much of the RDM correspondence between a given fMRI location and a given MEG time point could be attributed to each process of interest, as captured by the model RDMs (*Figure 3B*). Analogous to the MEG data, we computed RDMs (dimensions: 40×40) based on the voxel-wise BOLD activity patterns in 25 regions of interest (ROIs) (*Figure 3C*). Four ROIs in the early visual cortices (bilateral V1, V2, V3, and V4) and four bilateral regions in the ventral temporal cortices (VTC) selective to faces, houses, animals, and man-made objects were defined using independent retinotopy and object category functional localizers, respectively. ROIs in the frontoparietal cortices were defined based on differences in activation magnitudes between recognized and unrecognized trials (*Levinson et al., 2021*) and covered nine unilateral regions in the task-positive network (TPN); and eight unilateral or midline regions in the default mode network (DMN). As shown in *Levinson et al., 2021*, TPN regions had higher activation in recognized trials while DMN regions had stronger deactivation in recognized trials.

Relating an RDM derived from a given fMRI location to RDMs derived from all MEG time points yielded a time course of MEG-fMRI fusion (quantified as the squared Spearman's rho) for that ROI (*Figure 3—figure supplements 1–3*). For each ROI, we then conducted a commonality analysis to determine the portion of the shared variance between its RDM (from fMRI) and the MEG RDM at each time point that can be accounted for by each model RDM. This procedure yielded a time course of commonality coefficients for each ROI and each model RDM, indicating the temporal evolution of recognition-related information specific to the modeled process in that ROI. V4 was excluded from the analysis because its RDM did not have significant positive correlations with MEG RDMs, leaving 24 ROIs in the analysis presented below.

## Early, concurrent onsets of recognition-related information across large-scale brain networks

The results of the model-driven MEG-fMRI fusion are displayed in *Figure 4*. Significant effects related to the recognition model were evident in all ROIs (*Figure 4A*, left). The earliest significance onset was at 110 ms concurrently in multiple VTC (face, animal, and house), TPN (bilateral MFG, bilateral IPS, and right IFJ), and DMN (bilateral AG, left SFG) regions, all leading to a transient peak at ~150 ms (*Figure 4*, blue and *Figure 4—figure supplements 1–3*). This initial peak was followed by a dip and a cascade of the reestablishment of recognition model effects in TPN (190 ms onward), DMN regions (202.5 ms onward), and VTC regions (220 ms onward). In many of the TPN and DMN regions, the recognition model-related effects stayed significant at most time points during the 2 s period investigated. In contrast, recognition model-related effects in the VTC predominantly existed in the first 1000 ms but were largely undetectable from 1000 ms onward, and were characterized by a transient, intermittent occurrence (*Figure 4A*, left; *Figure 4B*, 'face' ROI; *Figure 4—figure supplement 1*), potentially driven by feedback activity from frontoparietal regions. Notably, recognition model-related effects in the EVC were late and transient; they became statistically significant at ~600 ms and disappeared by 1000 ms after stimulus onset.

Interestingly, the early, two-wave dynamics of recognition model-related effects did not occur uniformly across all ROIs within the same network. For instance, recognition model-related effects emerged relatively late in OFC and bilateral insulae (225–237.5 ms, *Figure 4* and *Figure 4—figure supplement 2*) compared to other TPN regions. Moreover, these three regions did not show the sustained, elevated effects observed in other TPN regions. Instead, their dynamics were characterized by continuous fluctuations, which resembled those observed in the VTC regions, consistent with the idea that these regions might have an especially close dialogue with the VTC (*Bar et al., 2006*; *Huang et al., 2021*).

We next sought to identify neural dynamics driven by the two-state model. Like the recognition model, we found significant two-state model activity in all ROIs (*Figure 4A*, right). Similarly, we also observed early, two-wave dynamics in multiple ROIs, with two peaks occurring shortly after the initial

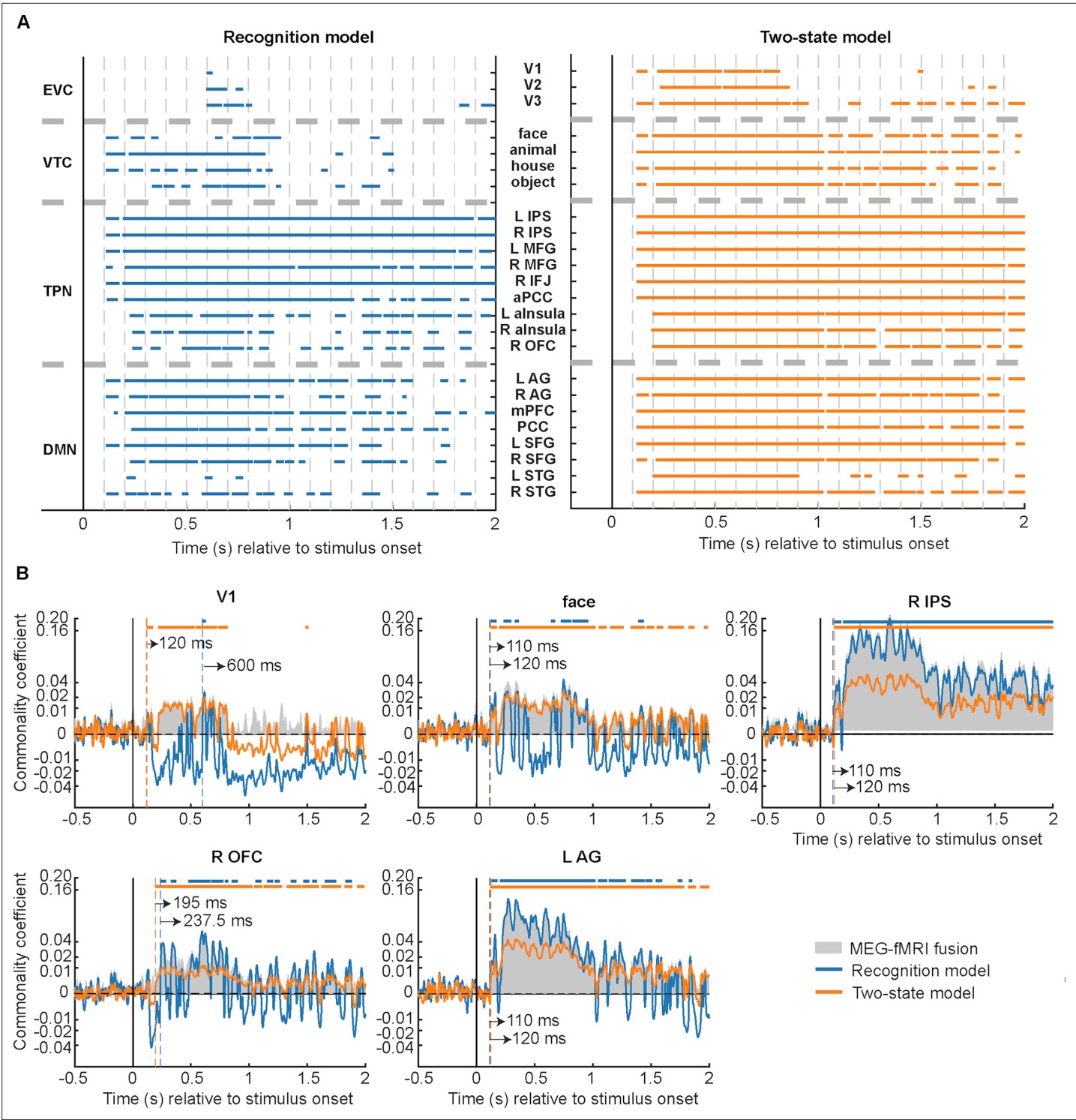

**Figure 4.** Model-based MEG-fMRI fusion results. (**A**) Commonality analysis results for all regions of interest (ROIs) are shown by plotting the statistically significant time points (p<0.05, cluster-based label permutation tests). Left: Results based on the recognition model. Right: Results based on the two-state model. (**B**) Commonality coefficient time course for recognition model (blue) and two-state model (orange) effects in five selected ROIs. The vertical dashed lines denote onset latencies, and the horizontal bars at the top indicate statistically significant time points (p<0.05, cluster-based label permutation tests). Gray-shaded areas denote the MEG-fMRI fusion, equivalent to the total variance shared by MEG and fMRI representational dissimilarity matrices (RDMs). Results from all ROIs are shown in *Figure 4—figure supplements 1–3*.

The online version of this article includes the following figure supplement(s) for figure 4:

*Figure 4 continued on next page*

**Figure supplement 1.** Model-based MEG-fMRI fusion results for early visual cortex (EVC) and ventral temporal cortex (VTC) regions of interest (ROIs).

**Figure supplement 2.** Model-based MEG-fMRI fusion results for all task-positive network (TPN) regions of interest (ROIs).

**Figure supplement 3.** Model-based MEG-fMRI fusion results for all default mode network (DMN) regions of interest (ROIs).

onset latency at 120 ms. There are, however, noticeable differences between the results derived from these two models. For instance, while EVC regions had by far the latest onset time for recognition model effects (600 ms), they were among the ROIs wherein the two-state model-related effects had the shortest onset latency (120 ms, *Figure 4B*, 'V1'; *Figure 4—figure supplement 1*). Moreover, in stark contrast to the transient nature of recognition model effects in EVC, the two-state model-related effects were sustained within the first second after stimulus onset. Overall, two-state model-related effects showed a more stable temporal dynamic than recognition model-related effects. This difference was most prominent in the VTC regions, but similar differences were also observable in the OFC and bilateral insulae.

In sum, both recognition and two-state models are associated with early, concurrent onset of significant processes across widespread ROIs, but the two models capture different aspects of recognition-related processing and are associated with distinct network-level patterns (e.g. two-state model accounts for EVC activity better while recognition model accounts for TPN activity better; see *Figure 4B* and *Figure 4—figure supplements 1–3*) and dynamic characteristics (e.g. two-state model-related effects are more sustained over time than recognition model-related effects). Below, we systematically characterize these network-level differences.

## Differential representational formats between functional brain networks

To assess network-level effects, we first examined the functional network association of each ROI, using well-known resting-state network (RSN) parcellation (*Yeo et al., 2011*). Because our TPN ROIs encompassed regions outside the occipitotemporal visual cortices that exhibited heighted activation in recognized than unrecognized trials (*Levinson et al., 2021*), they belonged to several RSNs. Therefore, these ROIs were assigned to the frontoparietal control network (FPCN, including the aPCC, bilateral MFG, and OFC), dorsal attentional network (DAN, including the right IFJ and bilateral IPS), and salience network (SAL, including bilateral insulae), according to the RSN that accounted for the highest percentage of voxels in each ROI (*Table 1*). For DMN ROIs, we excluded bilateral STG because >50% of their voxels were located in the somatosensory network based on Yeo et al. parcellation, leaving six ROIs in the DMN in the following analysis (bilateral AG, mPFC, PCC, and bilateral SFG). The extents of EVC and VTC were unchanged.

Visual inspection of commonality coefficient time courses (*Figure 4*, *Figure 4—figure supplements 1–3*) suggests that functional networks differed in the predominance of model-related effects. We thus estimated the explanatory powers of each model RDM by computing how much of the shared variance between MEG and fMRI RDMs it can explain. This was conducted for each ROI at each time point in the post-stimulus epoch. We were further interested in how the explanatory power of RDM models changed as a function of time. To this end, we split the explanatory power estimates into an early (0–1000 ms following stimulus onset) and a late (1000–2000 ms) time window, reasoning that perceptual processing had relatively strong influences on brain responses in the early stage whereas those in the late stage were mainly involved in post-perceptual processing.

To quantify the effect of model (recognition vs. two-state) and time window (early vs. late) on the amount of explained MEG-fMRI covariance, we conducted a two-way mixed-design analysis of variance (ANOVA). This analysis revealed a significant main effect of model in all brain networks, with opposite effects between ventral visual and higher-order frontoparietal networks (*Figure 5*). The two-state model was the superior model for explaining neural dynamics in EVC ($F_{1,799} = 1021.34$, p=$5.14 \times 10^{-145}$, $\eta^2_G$=0.32; *Figure 5A*) and VTC ($F_{1,799} = 892.66$, p=$2.79 \times 10^{-132}$, $\eta^2_G$=0.31; *Figure 5B*), while recognition model was superior for SAL ($F_{1,799} = 12.33$, p=$4.72 \times 10^{-4}$, $\eta^2_G$=0.01; *Figure 5C*), DAN ($F_{1,799}$ = 5104.20, p=0, $\eta^2_G$=0.77; *Figure 5D*), FPCN ($F_{1,799} = 740.60$, p=$6.42 \times 10^{-116}$, $\eta^2_G$=0.36; *Figure 5E*), and DMN ($F_{1,799} = 34.89$, p=$5.17 \times 10^{-9}$, $\eta^2_G$=0.03; *Figure 5F*).

**Table 1.** Distribution of each regions of interest (ROI's) voxels across different functional brain networks (as defined in *Yeo et al., 2011*), presented as percentage of voxels located in each network.

Each ROI was assigned to the brain network with the highest voxel count.

| | Visual | Somato | SAL | DAN | Limbic | FCPN | DMN |
|---|---|---|---|---|---|---|---|
| **SAL** | | | | | | | |
| L aInsula | 0.00 | 0.00 | 65.17 | 0.00 | 0.00 | 24.38 | 3.09 |
| R aInsula | 0.00 | 0.00 | 60.02 | 0.00 | 0.00 | 30.59 | 0.07 |
| | | | | | | | |
| **DAN** | | | | | | | |
| R IFJ | 0.00 | 0.00 | 0.00 | 79.43 | 0.00 | 12.18 | 0.00 |
| L IPS | 4.09 | 2.80 | 0.00 | 39.69 | 0.00 | 34.45 | 4.10 |
| R IPS | 6.91 | 0.00 | 0.00 | 55.98 | 0.00 | 20.71 | 0.01 |
| | | | | | | | |
| **FCPN** | | | | | | | |
| L MFG | 0.00 | 0.48 | 8.50 | 18.05 | 0.00 | 54.07 | 1.55 |
| R MFG | 0.00 | 0.00 | 6.97 | 0.01 | 0.00 | 78.21 | 0.16 |
| R OFC | 0.00 | 0.00 | 31.08 | 0.00 | 2.94 | 45.46 | 2.22 |
| aPCC | 0.00 | 0.00 | 0.29 | 0.00 | 0.00 | 36.77 | 20.53 |
| | | | | | | | |
| **DMN** | | | | | | | |
| L AG | 6.96 | 0.00 | 1.11 | 17.67 | 0.00 | 1.65 | 67.83 |
| R AG | 0.43 | 8.78 | 7.06 | 15.91 | 0.00 | 0.20 | 64.95 |
| PCC | 5.68 | 0.00 | 0.00 | 0.00 | 0.00 | 0.18 | 88.36 |
| mPFC | 0.00 | 0.00 | 0.06 | 0.00 | 0.06 | 0.33 | 91.73 |
| L SFG | 0.00 | 0.00 | 0.00 | 0.00 | 0.00 | 2.16 | 80.88 |
| R SFG | 0.00 | 0.43 | 16.62 | 0.00 | 0.00 | 21.78 | 36.95 |
| | | | | | | | |
| *Somatosensory/motor network (excluded from the analysis)* | | | | | | | |
| L STG | 0.00 | 45.35 | 7.34 | 0.76 | 0.35 | 3.63 | 38.46 |
| R STG | 0.10 | 69.18 | 5.35 | 1.77 | 0.00 | 1.51 | 17.58 |

There was a significant interaction effect between model and time window in all networks except DAN. In EVC, the significant interaction was driven by a sharp decrease in two-state model's explanatory power over time ($F_{1,799}$ = 155.62, p=9.27 × $10^{-33}$, $\eta^2_G$=0.07). In the remaining networks, the interaction was driven by an overall *increase* in two-state model's explanatory power and a decrease in recognition model's explanatory power over time (VTC: $F_{1,799}$=286.46, p=3.78×$10^{-55}$, $\eta^2_G$=0.13; SAL: $F_{1,799}$=11.35, p=7.90×$10^{-4}$, $\eta^2_G$=0.01; FPCN: $F_{1,799}$=97.65, p=8.38×$10^{-22}$, $\eta^2_G$=0.07; DMN: $F_{1,799}$=316.67, p=6.31×$10^{-60}$, $\eta^2_G$=0.20). Interestingly, this interaction effect manifested as a temporal shift in the dominant model in the DMN, with the recognition model prevailing in the early time window (*Figure 5F*, Wilcoxon signed-rank test, N=801, W=6944, p=3.66×$10^{-46}$) and the two-state model prevailing in the late time window (N=800, W=23,937, p=1.56×$10^{-11}$), suggesting a flexible adaption in DMN's functional profile to different stages of object recognition processing.

Furthermore, we repeated the same analysis using different choices of time ranges (with 100 ms and 200 ms sliding windows). As shown in *Figure 5—figure supplements 1–2*, the results remained qualitatively similar, providing additional empirical support for the robustness of our results.

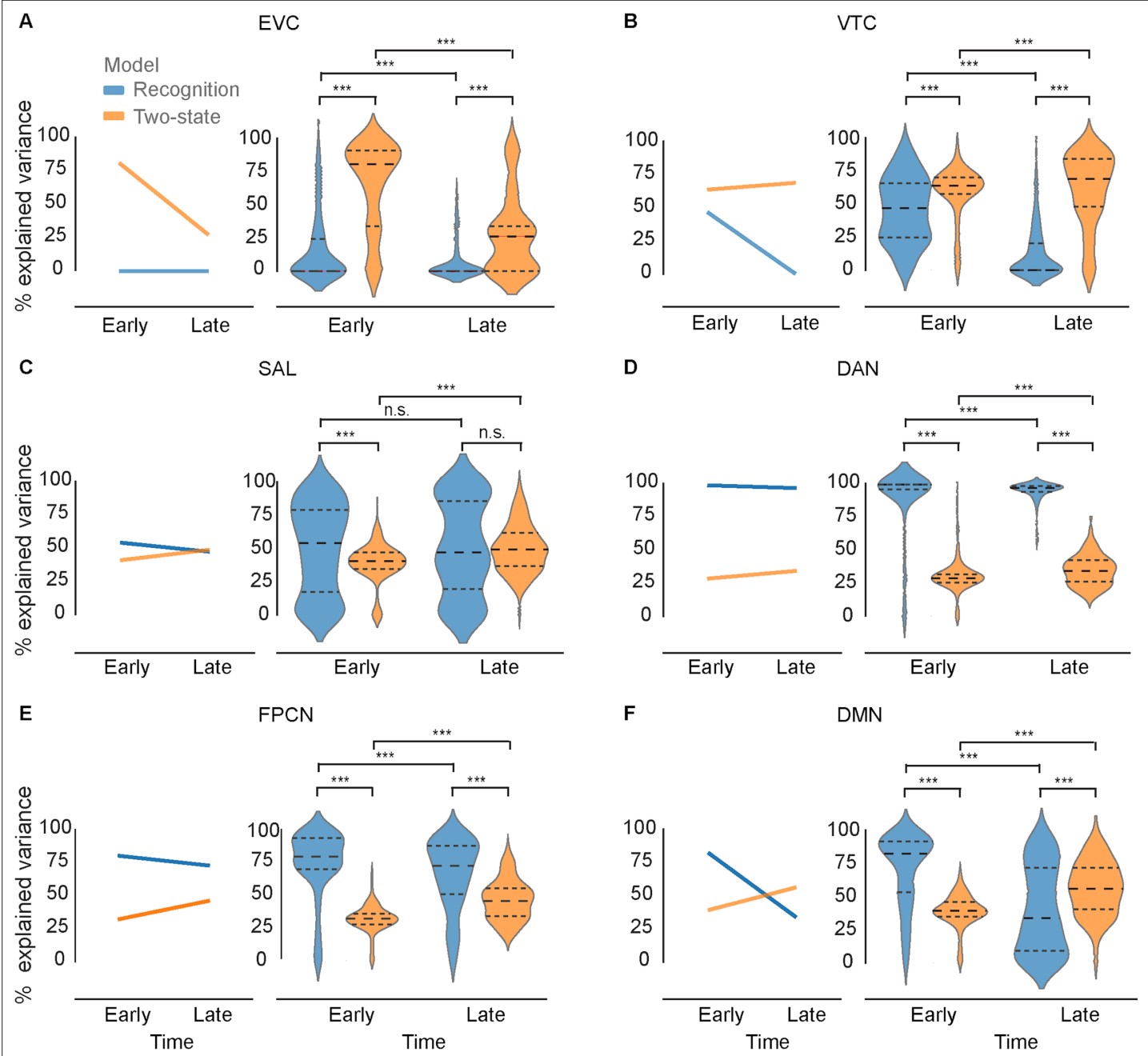

**Figure 5.** Percentage of shared variance between MEG and fMRI representational dissimilarity matrices (RDMs) explained by the recognition model (blue) and two-stage model (orange) in individual brain networks and two different post-stimulus time windows (early: 0–1000 ms; late: 1000–2000 ms). In all panels: Line plots depict the median percentage of explained variance across all time points in the window. The dashed lines in the violin plots represent 25th percentile, median, and 75th percentile. Asterisks indicate significant differences between time windows (early vs. late, two-sided Mann-Whitney test) or models (recognition vs. two-state, two-sided Wilcoxon signed-rank test) at p<0.001, Bonferroni-corrected across all pairwise tests. n.s.: not significant at a level of p<0.05, Bonferroni-corrected. EVC: early visual cortex, VTC: ventral temporal cortex, SAL: salience network, DAN: dorsal attentional network, FPCN: frontoparietal control network, DMN: default mode network. Results for individual regions of interest (ROIs) are presented in *Figure 5—figure supplement 3*.

The online version of this article includes the following figure supplement(s) for figure 5:

**Figure supplement 1.** Percentage of shared variance between MEG and fMRI representational dissimilarity matrices (RDMs) explained by the recognition model (blue) or the two-stage model (orange) in individual brain networks computed using 200 ms sliding windows in the post-stimulus period.

**Figure supplement 2.** Same as in *Figure 5—figure supplement 1*, with 100 ms sliding time windows.

*Figure 5 continued on next page*

*Figure 5 continued*

**Figure supplement 3.** Each model representational dissimilarity matrix (RDM's) explanatory power in individual regions of interest (ROIs), grouped according to their brain network membership and the time window after stimulus onset (early vs. late).

Together, these results reveal differential recognition-related representational dynamics between ventral visual and frontoparietal regions: Neural dynamics in the EVC and VTC were dominated by the two-state model effects, while the recognition model effects predominated in all other networks. In addition, compared to other networks, the salience (SAL) network had a relatively small main effect of model. Considering that SAL is situated between ventral visual and frontoparietal regions within the cortical hierarchy (*Margulies et al., 2016*), these results imply a gradual change in the representational format along the cortical hierarchy, similar to that observed previously in a different visual task (*González-García et al., 2018*). Importantly, the overall high percentages of the explained MEG-fMRI variance suggest that our choices of model RDMs were well suited for explaining MEG-fMRI fusion.

## Discussion

In the present study, we sought to provide a spatiotemporally resolved understanding of neural activity underlying object recognition under high uncertainty/low visibility conditions. To this end, we applied a model-based MEG-fMRI fusion approach to 7T fMRI data and MEG data collected using a threshold-level visual recognition task. Our results show that recognition-related information emerged simultaneously across multiple brain networks at 110 ms, but the representation format differed significantly between ventral visual and frontoparietal regions. While neural activity in visual regions is best captured by a state-switch process whereby recognized and unrecognized trials exhibit distinct activity patterns, neural activity in higher-order frontoparietal networks (including salience, dorsal attention, frontoparietal control, and default mode networks) are best described by a variability reduction process whereby recognized trials occupied a smaller region of the neural state space, yet with sharper representation of image category information (*Figure 6*). Together, these results reveal rich and heterogenous neural dynamics supporting object recognition under increased uncertainty. Below, we discuss the details and implications of these findings.

One common finding shared by the commonality analyses using different model RDMs was the early, parallel onsets (at 110 ms for the recognition model and 120 ms for the two-stage model) of recognition-related processes across multiple frontoparietal and ventral visual regions. The early onset observed in the higher-order frontoparietal regions is in accordance with previous MEG and primate neurophysiology findings, suggesting long-range recurrent processing between higher-order associative and ventral visual regions at an early stage during object recognition under challenging

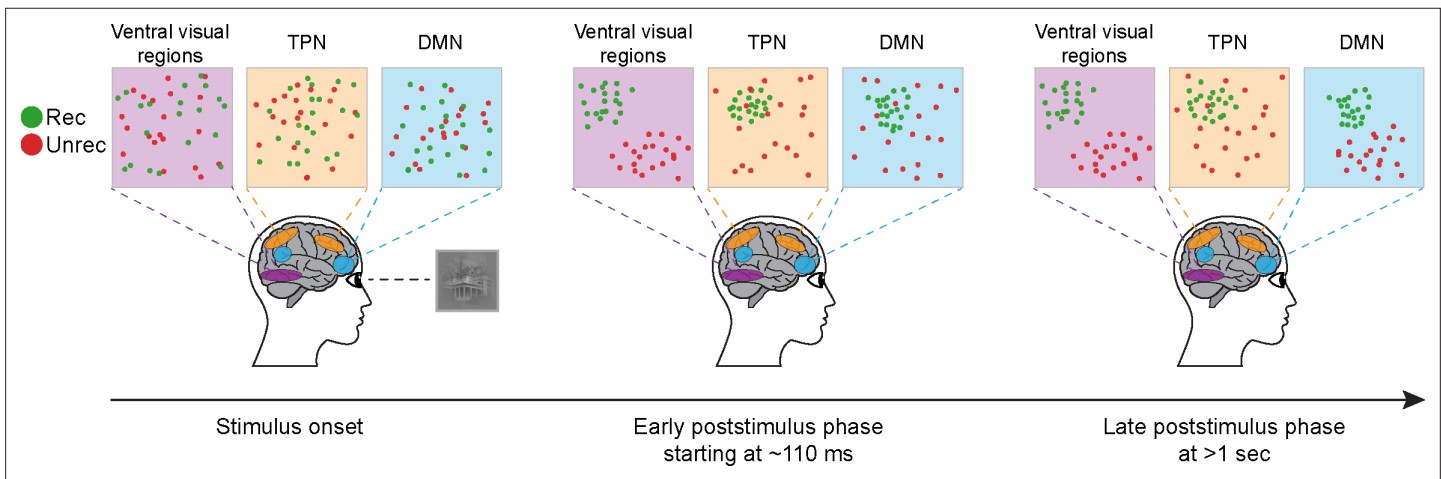

**Figure 6.** Schematic illustration of the changes in representational geometry across functional brain networks and time. Purple, orange, and blue colors rendered on the brain indicate the ventral visual regions, task-positive network (TPN), and default mode network (DMN). Squares in the upper panel display two-dimensional representational spaces in the corresponding brain areas, respectively. Green and red circles correspond to the representations of individual recognized and unrecognized image exemplars, respectively.

viewing conditions (*Bar et al., 2006*; *Kar and DiCarlo, 2021*; *Bellet et al., 2022*). The observation of parallel onsets across multiple cortical regions is somewhat surprising and requires a mechanistic explanation, but similar findings were previously reported in intracranial recordings in both primates and humans (*Barbeau et al., 2008*; *Siegel et al., 2015*; *Regev et al., 2018*). For instance, consistent with our results, Barbeau et al. showed that face-related signals onset simultaneously in fusiform gyrus and inferior frontal gyrus at 110 ms. These findings contrast with models positing a simple feedforward processing of visual inputs from primary sensory to high-order associative cortices. Instead, the simultaneity hints at the possibility of parallel processing of visual inputs across multiple cortical regions. For example, it has been suggested that the early frontoparietal recognition-related signals are triggered by the fast magnocellular inputs via the dorsal visual stream or projected from the pulvinar or mediodorsal nucleus of the thalamus (*Bullier, 2001*; *Bar, 2003*; *Kveraga et al., 2007*). The recognition-related signals in the ventral visual regions, on the contrary, are likely to be triggered by inputs via the parvocellular pathway along the ventral visual stream, which may be facilitated by feedback signals from dorsal stream regions (*Bar et al., 2006*).

The finding of differential model effects across large-scale functional networks suggests that recognition-related information in these networks differs in its representational format. Recognition-related information in visual regions (including EVC and VTC) is best characterized by a two-state representational format whereby activity patterns associated with both recognized and unrecognized exemplars clustered in two separated, relatively constrained representational subspaces. In contrast, recognition-related information carried by higher-order frontoparietal networks (including SAL, DAN, FPCN, and DMN) was dominated by the recognition model-related effects. They are linked with a representational format whereby activity patterns associated with recognized exemplars were clustered within a confined representational subspace, while activity patterns associated with unrecognized exemplars are distributed across a larger subspace. Importantly, despite the more confined representational subspace for recognized trials, category-level information is only present in recognized but not recognized trials (as shown by the decoding analysis), suggesting that the variability reduction in recognized trials might serve to reduce noise and facilitate the emergence of information related to the recognized stimulus content.

Previous monkey and human work associated a reduced neural variability with cortical engagement in stimulus and task processing (*Luczak et al., 2009*; *Ratcliff et al., 2009*; *Churchland et al., 2010*; *He, 2013*; *He and Zempel, 2013*; *Schurger et al., 2015*; *Arazi et al., 2017*). In line with this interpretation, our finding of reduced neural variability in higher-order frontoparietal regions during recognized trials suggests that successful recognition is linked with enhanced stimulus processing in these regions (consistent with the emergence of category-level information mentioned above). On the other hand, the high neural variability in unrecognized trials suggests that failed recognition may partially result from limited frontoparietal engagement, at least in the context of challenging viewing conditions. Importantly, the unbalanced stimulus processing between recognized and unrecognized trials were limited to higher-order frontoparietal regions; EVC and VTC neural dynamics revealed a symmetrical, two-state representational format.

The asymmetrical reduction in neural variability was particularly strong in the FPCN and DAN, explaining 80% and 98.4% of total shared variance between fMRI and MEG RDMs in the 0–1000 ms time window. These areas are sources of attentional and executive control (*Miller and Cohen, 2001*; *Corbetta and Shulman, 2002*; *Duncan, 2010*; *Ptak, 2012*; *Buschman and Kastner, 2015*). Accordingly, it is possible that the enhanced FCPN and DAN engagement during recognized trials reflect attention-related modulations that support object recognition (*Serences et al., 2004*; *Baldauf and Desimone, 2014*). In light of the increasing evidence that FCPN and DAN regions are actively involved in object recognition processing (*Konen and Kastner, 2008*; *Freud et al., 2017*; *Levinson et al., 2021*; *Ayzenberg and Behrmann, 2022*; *Mei et al., 2022*), the enhanced FCPN and DAN engagement may be directly related to perceptual processing per se. For example, the early recognition-related responses in the FPCN and DAN may reflect an initial, fast analysis of noisy visual input in the dorsal visual stream which facilitates fine-grained analysis happening in the ventral stream (*Bullier, 2001*; *Bar, 2003*; *Bar et al., 2006*). Alternatively, they may reflect the accumulation of sensory evidence preceding the rise of a particular perceptual experience (*Gold and Shadlen, 2007*; *Kelly and O'Connell, 2015*). These interpretations are not mutually exclusive and consistent with our empirical finding that the onset of recognition-related activity according to both models

(significance at 110–120 ms) is earlier than the onset of category-level information (initial peak at 290 ms, significance at 470 ms).

Using whole-brain MEG decoding, we found that object category-related information reached significance at 470 ms. This is markedly later than those reported in previous studies using high-contrast, clear object images, where category decoding typically peaked within the first 200 ms (*DiCarlo et al., 2012*; *Carlson et al., 2013*; *Cichy et al., 2014*; *Isik et al., 2014*). Our control task and analysis confirmed that this latency difference was driven by a difference in stimulus visibility/uncertainty: In a localizer task (with the same image set as used in the main task but presented at suprathreshold contrasts), we found object category decoding reached significance at 110 ms and peaked at 170 ms. These results are consistent with the idea that more time-consuming recurrent neural processing is needed for object recognition under high uncertainty/low visibility. This interpretation is also consistent with recent human electrophysiology findings showing that the latency for the emergence of object category information increases as the viewing duration becomes shorter (*Mohsenzadeh et al., 2018*) or the visible part of an object shrinks (*Tang et al., 2018*; *Rajaei et al., 2019*) and fits with the relatively long behavioral responses (up to >1 s) for the categorization of degraded visual images (*Philiastides and Sajda, 2007*; *Ratcliff et al., 2009*; *Hegdé and Kersten, 2010*; *Philiastides et al., 2011*).

Finally, our study adds to a growing body of evidence showing the involvement of DMN in visual perceptual processing (*González-García et al., 2018*; *González-García and He, 2021*; *Levinson et al., 2021*; *Smallwood et al., 2021*). For instance, a previous study using the 7T fMRI dataset investigated herein reported that despite stronger deactivation in recognized trials, DMN regions encoded the content of recognition within its (deactivated) activity patterns (*Levinson et al., 2021*). Here, using stimulus-induced reduction in neural variability as an indicator of cortical engagement, we found that successful recognition is associated with an enhanced DMN engagement despite its deactivation, and that neural activity in the DMN dynamically shifts from recognition model-dominated effects (variability reduction in recognized trials) to two-state model-dominated effects (bifurcation between recognized and unrecognized trials). The shift toward the two-state representational structure in the late stage may indicate an enhanced intercommunication with other areas that were also dominated by the two-state model effects, such as the ventral visual regions. Speculatively, given DMN's prominent role in both associative processing (*Bar et al., 2007*; *Stawarczyk et al., 2021*) and prior-guided perception (*González-García et al., 2018*; *Flounders et al., 2019*), this effect might reflect interactions between the current visual inputs and the observer's prior knowledge that guide subsequent perceptual decisions and related associations.

In summary, under a challenging viewing condition, we found that object recognition-related neural processes emerge early (110–120 ms) and simultaneously across multiple ventral visual and frontoparietal regions despite the late emergence (470 ms) of category-related information. Importantly, brain regions differ in the temporal evolution and representational format of their neural dynamics, in a manner that is roughly organized according to large-scale functional networks. Thus, object recognition under uncertainty engages a bidirectional coordination between multiple cortical networks with diverse functional roles at an early processing stage.

## Methods
### Participants

Twenty-five volunteers (15 females, mean age 26 years, range 22–34 years) participated in the MEG experiment, approved by the Institutional Review Board of the National Institute of Neurological Disorders and Stroke (protocol #14N-0002). One participant did not complete the MEG experiment due to discomfort and was removed from further procedures, leaving 24 participants in the MEG dataset. A total of 38 volunteers (26 females, mean age 27 years, range 20–38 years) participated in the fMRI experiment, following the study protocol (s15-01323) approved by the Institutional Board Review of New York University School of Medicine. Ten volunteers were excluded due to not completing the task and three were removed from data analysis due to poor behavioral performance, leaving 25 volunteers in the fMRI dataset. All participants were right-handed, neurologically healthy, and had a normal or corrected-to-normal vision. All experiments were conducted in accordance with the Declaration of Helsinki and written informed consent was obtained from each participant. Both

the MEG and fMRI datasets analyzed herein have been used in previous publications (*Podvalny et al., 2019*; *Levinson et al., 2021*).

## Experimental stimuli and procedure

The same stimuli set was used in both the MEG and fMRI experiments. The stimuli set consisted of five unique exemplars from four categories, namely face, house, man-made object, and animal. These images were selected from public domain labeled photographs or from Psychological Image Collection at Stirling (PICS, http://pics.psych.stir.co.uk). The images were resized to 300×300 pixels and converted to grayscale. In addition, the pixel intensities were normalized by subtracting the mean and dividing by the standard deviation, and image was filtered with a two-dimensional Gaussian kernel with a standard deviation of 1.5 pixels and 7×7 pixels size. We additionally generated scrambled images by shuffling the two-dimensional Fourier transformed phase of one randomly chosen exemplar from each category (data from scrambled image trials were not used in the present study due to low trial count). This yielded a stimuli set consisting of 20 unique real images and 4 scrambled images in total. In both MEG and fMRI experiment, stimulus size was ~8 degrees in diameter. Prior to the main task, participants underwent an adaptive staircase procedure 'QUEST' to identify image contrast yielding a recognition rate of 50%. Detailed procedures are available in *Podvalny et al., 2019*; *Levinson et al., 2021*.

In the main task, each trial started with a central fixation cross on a gray background for a variable duration (MEG: 2–4 s, fMRI: 6–20 s). This was followed by image presentation at central fixation (with a visible fixation cross) for 66.7 ms, during which the image intensity gradually increased from 0.01 to the threshold contrast intensity. After a variable delay period (MEG: 2–4 s, fMRI: 4 or 6 s) wherein the screen returned to the initial gray background with a fixation cross at the center, participants were prompted to answer two questions. The first was a four-alternative-forced-choice task, wherein participants reported the object category of the presented image with a button press. Participants were instructed to make a genuine guess if they did not recognize the object. The stimulus response mapping, indicated by the order of words 'face'/'house'/'object'/'animal' on the screen, was randomized across trials. The second question asked about participants' recognition experience. They were instructed to report 'yes' if they saw a meaningful content, and 'no' if they saw nothing or random noise patterns. In the MEG experiment, participants had up to 3 s to indicate their response to each question and the next screen was presented whenever a response was given, while in the fMRI experiment each question was presented for a fixed 2 s period.

Both the MEG and fMRI main tasks consisted of 300 real image and 60 scrambled image trials. Each unique image was presented 15 times throughout the main task. In the MEG experiment, trials were randomized and split into 10 experimental runs. In the fMRI experiment, trials were split into 15 experimental runs, each with the entire stimulus set presented once in a random order.

Participants in both the fMRI and MEG experiments additionally completed a 'localizer' task. The localizer task in the fMRI experiment was described in detail in *Levinson et al., 2021*, and the data were used for the definition of VTC (face, house, object, animal) ROIs used in that paper and in the current study (for detailed method about localizer data analysis, see *Levinson et al., 2021*). Here, we describe in detail the methods for the localizer task in the MEG experiment, as the associated data have not been previously published. In the localizer task block, participant viewed the same set of 20 real images as used in the main task (scrambled images were not included), with the crucial difference being that the images were presented at the original high contrast. This allowed us to directly compare the neural dynamics of object recognition triggered by the same set of images at different visibility/uncertainty levels. The localizer block consisted of 300 trials, with each unique image presented 15 times in a randomized order. Each trial lasted 1 s and started with an image presentation of 66.7 ms, during which the image intensity gradually increased from 0.01 to 1 (the original contrast). Participants were instructed to perform a one-back memory task and press a button whenever they saw an image presented twice consecutively.

## MEG data acquisition and preprocessing

Continuous MEG signals were recorded at a sampling rate of 1200 Hz using a 275-channel whole-head MEG system (CTF, VSM MedTech). Three electrodes were dysfunctional and excluded, leaving 272 channels for further procedures. MEG data were preprocessed with the MNE toolbox, Python

(*Gramfort et al., 2013*). Independent component analysis (ICA) was performed on continuous data from each run to remove artifacts related to eye movements, blinks, and heartbeats. The ICA-cleaned MEG data were detrended, demeaned, band-pass filtered between DC and 35 Hz, and down-sampled to 400 Hz (for commonality analysis) or 100 Hz (for decoding analysis). We epoched continuous MEG data into 2.5 s trials from –500 ms to 2000 ms relative to stimulus onset and applied baseline correction for each sensor using the pre-stimulus time window.

## Multivariate pattern analysis of MEG data

Decoding on MEG data was performed using The Decoding Toolbox (*Hebart et al., 2014*). For each participant, the decoding analysis was performed using linear SVM (in LIBSVM implementation, *Chang and Lin, 2011*) with a fixed cost parameter of $c$=1. Data associated with scrambled images were excluded from the analysis, leaving data associated with 40 conditions (5 unique real exemplars×4 categories×2 recognition outcomes).

For each time point (from –500 ms to 2000 ms relative to stimulus onset, in 10 ms increments), preprocessed MEG data derived from every single trial were arranged as a 272-dimensional vector (corresponding to 272 sensor responses) and served as a sample in the classification models. To decode recognition outcomes from MEG signals at a given time point, we employed a fivefold cross-validation scheme. We pseudo-randomly divided data into five subsets, each containing approximately the same percentage of samples from each condition as the complete dataset. We iteratively trained SVM classifiers to discriminate between samples from different conditions using data from four folds as the training set and tested for their generalization performance on the samples from the remaining fold. To minimize the potential classifier bias due to the unequal distribution of conditions in the training set, we employed a resampling method. For each cross-validation iteration, we built up to 100 different classification models, each trained on a random subset of samples from the more frequent class matched to the sample size of the less frequent class in the training set. The label predicted by the majority of the models was then selected as the predicted label for a particular test sample. This training test process was repeated until all folds had been used as the test set once. We computed balanced decoding accuracy across all cross-validation folds to evaluate the prediction performance. Balanced decoding accuracy of 50% indicates no predictive power, whereas a score of 100% indicates perfect predictive power.

An analogous procedure was employed to decode object category in recognized and unrecognized trials, respectively. For each time point, we performed the aforementioned binary classification strategy on all six possible pairwise combinations of the four object categories. Decoding scores derived from all pairwise classifications were then averaged to yield an estimate of the overall decoding score across all pairwise classifications. For the object category decoding in the localizer task block, we employed a similar procedure, but without the resampling procedure described above because the number of samples was already balanced across the four categories (75 each).

The group-level statistical significance for the balanced decoding accuracy at each time point was estimated using a one-tailed Wilcoxon signed-rank test against chance level (50%). To control for multiple comparisons, we performed cluster-based inference (*Maris and Oostenveld, 2007*), whereby a cluster was defined as contiguous time points showing significant group-level decoding accuracy at a $p<0.05$, uncorrected level (i.e. cluster-defining threshold), and a cluster-level statistic was computed by taking the sum of the $W$ statistics of the Wilcoxon signed-rank test across all time points in a given cluster. The statistical significance of a given cluster was determined using a sign permutation test that randomly flipped the sign of subjects' data (i.e. decoding accuracy minus chance) with 50% probability, 5000 times (*Nichols and Holmes, 2002*; *Martin Cichy et al., 2017*; *Hebart et al., 2018*). For each permutation sample, we computed the $W$ statistic for each time point and took the largest of the cluster-level statistics, resulting in an empirical distribution of maximum cluster-level statistics. We reported a cluster in the original data as significant if its cluster-level statistic exceeded the 95th percentile of the empirical distribution of maximum cluster-level statistics (corresponding to $p<0.05$, cluster-corrected).

To construct an RDM for each time point (from –500 to 2000 ms relative to stimulus onset, in 2.5 ms increments) and each participant, we averaged the whole-brain sensor-level activity patterns for each of the 40 conditions and computed 1-Pearson's correlation between each condition pair as the dissimilarity measure. Dissimilarities were then assembled in a 40×40 matrix. Dissimilarities

between recognized exemplars were stored in the upper-left quadrant, and those between unrecognized exemplars were stored in the bottom-right quadrant. Values in the bottom-left and upper-right quadrants represented dissimilarities between recognized and unrecognized exemplars. The resulting RDM was symmetric around the diagonal, with the diagonal being undefined. We averaged RDMs across participants (N=24), yielding one RDM for each time point. To facilitate the visualization of relations between conditions, the mean RDMs at selected time points were projected onto a two-dimensional plot using MDS (*Kruskal and Wish, 1978*; *Shepard, 1980*).

We constructed a category model RDM to test for category-related processing in recognized and unrecognized image trials, respectively. The category model RDM had a size of 20×20 cells, corresponding to the total number of image exemplars for a given recognition outcome. It predicted that activity patterns evoked by images from the same category would have relatively small dissimilarities with each other, while dissimilarities between activity patterns evoked by different categories would be relatively large. For each participant, we compared the category model RDM with the parts of their MEG RDMs representing recognized and unrecognized trials (the upper-left and bottom-right quadrants), respectively. That is, we computed the Spearman's rank correlation coefficient (rho) between the model RDM and MEG RDM derived from each time point. Statistical significance at the group-level was determined using procedures analogous to those employed for the decoding analysis. Spearman's rho significant above zero indicated the presence of category information.

## MRI data acquisition, preprocessing, and ROI definition

Detailed methods related to the fMRI dataset have been reported in *Levinson et al., 2021*. Here, we briefly summarize the procedures relevant to the present work. MRI data were acquired using a Siemens 7T scanner equipped with a 32-channel NOVA coil. A T1-weighted structural image was acquired using an MPRAGE sequence (TR=3000 ms, TE=4.49 ms, 192 saggital slices, flip angle=6, FOV=256×256 mm$^2$, 1.0 mm isotropic). Functional volumes covering the whole brain were obtained using T2* weighted echo planar imaging sequence (TR=2000 ms, TE=25 ms, FOV=192×192 mm$^2$, flip angle=50, acceleration factor/GRAPPA=2, multi-band factor 2). Each volume consisted of 54 oblique slices with an in-plane resolution of 2×2 mm and a slice thickness of 2 mm with a distance factor of 10%. fMRI data collected from each participant were preprocessed using FSL package (http://fsl.fmrib.ox.ac.uk/fsl/fslwiki/FSL). In short, they were realigned, corrected for slice timing, spatially smoothed with a 3 mm FWHM kernel, and registered to the T1 anatomical scan. Artifacts related to motion, arteries, or CSF pulsation were removed using ICA.

A retinotopy functional localizer was used to define EVC ROIs in each participant. For each hemisphere, field maps in V1, V2d, V2v, V3d, V3v, and V4 were identified, and the dorsal and ventral portions of V2 and V3 within each hemisphere were subsequently merged into a single ROI, respectively. For the present study, we further merged the left and right hemisphere for each early visual area, resulting in four EVC ROIs including the bilateral V1, V2, V3, and V4. EVC ROIs could not be defined in several individuals due to noisy data, leaving 23 subjects' data in analyses involving V1–V3 and 22 subjects' data in analyses involving V4. The four category-selective ROIs in the VTC (face, house, object, animal) were defined individually using a separate functional localizer: each ROI consisted of up to 500 most responsive voxels to one of the four object categories within occipitotemporal visual cortex (thresholded at Z>2.3, masked with the conjunction of inferior lateral occipital cortex, occipital fusiform gyrus, posterior parahippocampal gyrus, temporal fusiform cortex, and temporal occipital fusiform cortex from the Harvard-Oxford brain atlas). In addition, a category-selective ROI was only defined if more than 100 voxels surpassed the threshold within the mask. This yielded face-selective ROIs in 20 subjects, house-selective ROI in 15 subjects, object-selective ROI in 18 subjects, and animal-selective ROI in 18 subjects. TPN and DMN ROIs were defined based on significant clusters extracted from the group-level statistical maps for the recognized vs. unrecognized general linear model (GLM) contrast (cluster-corrected p<0.05). These clusters were transformed back to each individual's native space; thus, TPN and DMN ROIs were defined for all 25 participants.

## fMRI multivariate pattern analysis

We constructed a 40×40 RDM for each ROI based on the BOLD activity pattern elicited by each exemplar in each recognition condition. These activity patterns were estimated using a first-level GLM implemented in FSL. For each experimental run, we defined one regressor for each image exemplar

associated with either recognition outcome aligned to the stimulus onsets. An additional nuisance regressor, aligned to the onset of the first question, was added to account for motor-related activation. This resulted in a total of 49 regressors per run ((20 real exemplars+4 scrambled exemplars)×2 recognition outcomes+1 nuisance regressor), each convolved with a gamma-shaped hemodynamic response function. Note that each exemplar was only presented once in an experimental run and could be only associated with a specific recognition outcome. As a consequence, only 25 regressors contained non-zero values in each run. The same GLM was estimated for all experimental runs, yielding trial-wise beta parameter estimates for activity patterns elicited by each of the presented stimuli in each recognition condition. As for the MEG RDM, we discarded data related to the scrambled images, leaving data associated with 40 conditions (20 real exemplars×2 recognition outcomes). We averaged the activity patterns across runs for each condition and then computed the dissimilarities between all condition pairs, before entering them into the RDM. Lastly, RDMs were averaged across participants, yielding a mean 40×40 RDM for each ROI.

## Model-driven MEG-fMRI fusion

To provide a spatiotemporally resolved view of object recognition processing under uncertainty, we performed a model-driven MEG-fMRI fusion based on RSA (*Hebart et al., 2018*; *Flounders et al., 2019*). First, we applied cross-modal RSA to combine the MEG and fMRI data acquired from independent participant groups. For a given MEG time point and a given fMRI ROI, we extracted the lower triangles of the corresponding 40×40 RDMs, respectively, and converted them to dissimilarity vectors. We then calculated the squared Spearman' rank correlation coefficient between the MEG and fMRI dissimilarity vectors, which is equivalent to the variance shared by MEG and fMRI RDMs. Repeating this procedure for all MEG time points and fMRI ROIs, we obtained a time course of MEG-fMRI fusion that indexed the representational similarity between whole-brain MEG data and ROI-specific fMRI data. Since MEG and fMRI data were collected from different groups of participants, we conducted the described procedure using the group-average RDMs from each recording modality as the best estimate of the true RDM (*Hebart et al., 2018*).

To isolate the effects related to a neural process of interest in the MEG-fMRI fusion, we constructed model RDMs that reflected the expected dissimilarity pattern related to a neural process of interest. The contribution of the neural process of interest to the MEG-fMRI fusion at a given time point and ROI can be estimated using commonality analysis (*Seibold and McPHEE, 1979*; *Hebart et al., 2018*; *Flounders et al., 2019*). More specifically, a commonality coefficient *C* yielded from the analysis informed about how much of the shared variance between the MEG RDM at a given time point and the fMRI RDM from a given ROI can be attributed to the model RDM. Formally, the commonality coefficient at a given time point and a given location is calculated as follows:

$$C(MEG, fMRI, model) = R^2_{MEG, fMRI} + R^2_{MEG.model} - R^2_{MEG.fMRI,model}$$

We applied a cluster-based label permutation test to determine the statistical significance of commonality coefficients (*Hebart et al., 2018*; *Flounders et al., 2019*). For each of the 5000 permutations, we randomly shuffled the labels of MEG RDM, computed the commonality coefficient time course for a given model, and extracted the maximum cluster-level statistic. The cluster-defining threshold at each time point was set to the 95th percentile of the null distribution yielded from the permutation test (equivalent to p<0.05, one-tailed). Analogous to the statistical testing for MEG decoding, a cluster in the original data was considered significant if its cluster-level statistic exceeded the 95th percentile of the null distribution of maximum cluster-level statistics (corresponding to p<0.05 cluster-corrected).

## Analysis on RDM models' explanatory power

Model RDM's explanatory power at a given time point was estimated by commonality coefficient divided by the shared variance between MEG and fMRI RDM, which quantifies the percentage of the shared variance between MEG and fMRI RDM that can be accounted for by the model. The explanatory power estimate was set to zero if the commonality coefficient was negative. The estimation was conducted for all 801 time points between 0 and 2000 ms relative to stimulus onset, yielding a time course of percentages of explained MEG-fMRI covariance per ROI. The time course was split into two

halves, with the first half corresponding to 0–1000 ms after stimulus onset and the second half corresponding to 1001–2000 ms after stimulus onset.

To estimate each model's effect at the network level, we averaged time courses of percentages of explained MEG-fMRI covariance across ROIs within each network. For each network, we conducted a two-by-two mixed-design ANOVA, with model (recognition model vs. two-state model) as the repeated-measures factor and time (early vs. late) as the between-group factor, and individual data points being percentages of explained MEG-fMRI covariance at each time point. Post hoc tests between models were conducted using a two-sided Wilcoxon signed-rank test, while post hoc tests between times were carried out using a two-sided Mann-Whitney test. Bonferroni correction was applied to control for multiple comparisons across post hoc tests. We reported statistical significance at a $p < 0.05$, Bonferroni-corrected level.

## Acknowledgements

This work was supported by an NIH grant to BJH (R01EY032085). The authors would like to thank Richard Hardstone and Max Levinson for useful discussions.

## Additional information

### Funding

| Funder | Grant reference number | Author |
|---|---|---|
| National Institutes of Health | R01EY032085 | Biyu J He |

The funders had no role in study design, data collection and interpretation, or the decision to submit the work for publication.

### Author contributions

Yuan-hao Wu, Conceptualization, Formal analysis, Validation, Investigation, Visualization, Methodology, Writing – original draft, Writing – review and editing; Ella Podvalny, Data curation, Investigation, Writing – review and editing; Biyu J He, Conceptualization, Resources, Supervision, Funding acquisition, Writing – original draft, Project administration, Writing – review and editing

### Author ORCIDs

Yuan-hao Wu http://orcid.org/0000-0002-5631-5082
Biyu J He http://orcid.org/0000-0003-1549-1351

### Ethics

Human subjects: Data collection procedures followed protocols approved by the institutional review boards of the intramural research program of NINDS/NIH (protocol #14 N-0002) and NYU Grossman School of Medicine (protocol s15-01323). All participants provided written informed consent.

### Decision letter and Author response

Decision letter https://doi.org/10.7554/eLife.84797.sa1
Author response https://doi.org/10.7554/eLife.84797.sa2

## Additional files

### Supplementary files
• MDAR checklist

### Data availability

The analysis code, data and code to reproduce all figures can be downloaded at https://github.com/BiyuHeLab/eLife_Wu2023, (copy archived at *BiyuHeLab, 2023*).

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
