## [Editor Report]

This study presents valuable findings on the fine spatiotemporal profiles of object recognition in human brains under noisy and ambiguous conditions. The evidence supporting the conclusion is compelling, with state-of-the-art techniques and model-driven fusion of MEG and 7T fMRI. The work will be of broad interest to cognitive neuroscientists working on consciousness and related fields.

---

## [Decision Letter]

**Decision letter after peer review:**

Thank you for submitting your article "Spatiotemporal neural dynamics of object recognition under uncertainty in humans" for consideration by *eLife*. Your article has been reviewed by 2 peer reviewers, one of whom is a member of our Board of Reviewing Editors, and the evaluation has been overseen by Christian Büchel as the Senior Editor. The following individual involved in the review of your submission has agreed to reveal their identity: Zirui Huang (Reviewer #2).

The reviewers have discussed their reviews with one another, and the Reviewing Editor has drafted this to help you prepare a revised submission. Both reviewers acknowledge the importance of the questions addressed in the work and the state-of-the-art model-driven MEG-7T fusion analysis. In the meantime, the reviewer raised several crucial concerns about the validity of several methods and the underlying rationale, as well as the lack of statistical support for some conclusions.

Essential revisions:

(1) There are some confusions in the results, such as the inconsistencies between the decoding and RSA analysis, the strange outlier pattern that might confound the results, and the ROI selection that might bias model comparison. The authors should address the points by providing convincing and thorough evidence. (See comments by Reviewer 1)

(2) One of the key conclusions of the work, i.e., variability reduction, is not backed up by statistical evidence (reviewer 1).

(3) Several choices in the analysis seem arbitrary and lack verification, such as model comparison in different ROIs, and the use of two time ranges to test the hypothesis (see comments by Reviewer 1).

(4) Since 7T fMRI results provide detailed spatial information as well as early recognition activities that might arise through a subcortical pathway, it seems feasible and a valuable opportunity to delineate the underlying mechanism by including subcortical regions in the analysis (see Reviewer 2's comments).

*Reviewer #1 (Recommendations for the authors):*

1. In Figure 2C, The RSA results seem to be not consistent with the decoding results shown in Figure 2B. If my understanding is correct, we would expect to see category decoding starting at 290 ms, but the RSA results are a bit uniform and do not display any clustering effect as would be the case for successful category decoding. Why is that?

2. Moreover, what do the horizontal and vertical yellow lines indicate? Do they represent some specific image exemplars that elicit abnormal MEG responses? I am concerned that they indeed contribute to the weak category decoding for unrecognized trials given that they are within the unrecognized quadrate.

3. In Figure 2D, the shrinking neural representation space of unrecognizable objects does not have statistical support. The authors should do a quantitative analysis to back up the claim.

4. The authors chose ROIs based on the difference between recognized and unrecognized trials for further analysis, i.e., testing the recognition model vs. the two-state model. What is the rationale behind the pre-selection? Would such ROI selection criteria be biased toward the recognition model?

5. The idea of using a "recognition model" and "two-state model" to test the representational neural format in different ROIs is interesting but unfortunately somewhat unconvincing. For example, the only difference between the two models seems to reside in the 4th quadrant, that is, the category RSA for unrecognized trials. In other words, I don't think the two models are good comparisons. Related to this point, Figure 4A is confusing since the two models work for all ROIs.

6. Related to the above point, if the major goal is to characterize the temporal evolvement of representational models in different regions, the authors should compare the two models directly at each time point and in each ROI. This seems to be a more straightforward way.

7. The authors selected 0-1000ms and 1000-2000ms as the perceptual and post-perceptual time range for model comparison. This division is quite arbitrary and lacks independent criteria. Moreover, 1000 ms is too long for perceptual analysis and would certainly involve post-perceptual and feedback signals, in contrast to the perceptual claim.

8. The major findings based on commonality analysis rely on decoding results of both MEG and fMRI. Figure 2B shows that unrecognizable trials in MEG signals do not contain category information. But how about fMRI decoding results? Do the ventral visual pathway regions show successful category decoding for unrecognizable trials? If yes, could the two-state results for VTC be simply due to fMRI results?

*Reviewer #2 (Recommendations for the authors):*

Wu and colleagues applied a novel MEG-fMRI fusion approach and studied the spatiotemporally resolved neural activity of object recognition under uncertainty through a reanalysis of MEG and fMRI datasets from the same research group (Podvalny et al., 2019; Levinson et al., 2021). The authors found that recognition-related brain activity occurs prior to category-related activity, and demonstrated differential representation formats (i.e., state-switch vs. variability-reduction) in ventral visual and frontoparietal regions. It was concluded that object recognition under uncertainty engages bidirectional coordination among multiple cortical networks with diverse functional roles at an early processing stage. The manuscript was very well-written. I have a few points to share as follows.

1. Perhaps many readers are not familiar with the MEG-fMRI fusion approach. A few sentences may be needed to unpack the rationale and basic assumptions of this approach. For example, measurements of neural activity by different imaging modalities are presumed to reflect the same generators when they correlate on a trial-by-trial or condition-by-condition basis.

2. One intriguing finding is that the onset of recognition-related activity (110-120ms according to both models) occurs much earlier than the onset of category-related activity. I wonder if the authors could discuss more about the early-stage activity. Does it suggest that the participants were aware of the content of the image at this stage? Or the neural process of early-stage activity remains under an unconscious processing stage, and the actual recognition experience occurred later at ~500ms (according to the peak decoding of recognition outcome)? A possible interpretation might be that the recognition-related activity at 110-120ms is an unconscious process associated with an initial assessment of the quality of sensory evidence, which could be accumulated over time (e.g., through corticocortical feedback activity) until the emergence of recognition experience.

3. The authors mentioned that "the phase-scrambled images were not included in the analyses reported below due to comparatively low trial numbers". However, the scrambled conditions were analyzed by both Podvalny et al. 2019 and Levinson et al. 2021. Also considering that the MEG-fMRI fusion approach used in the current study was based on a condition-by-condition matrix instead of a trial-by-trial matrix, I wonder if false alarm vs. correct rejection (derived from scrambled conditions) would be information that is complementary to the current findings.

4. The early onset of frontoparietal recognition-related signals is of particular interest. The authors interpreted that they are "triggered by the fast magnocellular inputs via the dorsal visual stream or projected from the pulvinar or mediodorsal nucleus of the thalamus." As the current study used 7T-fMRI data with high spatial resolution, studying subcortical regions should be feasible. In addition, Levinson et al. 2021 identified differences in activation magnitudes between recognized and unrecognized trials in a few subcortical regions including the thalamus and brainstem. It might be worth including these subcortical regions in the ROIs.

5. It was found that successful recognition was associated with enhanced DMN engagement despite its deactivation. There were dynamic shifts from recognition model-dominated effects to two-state model-dominated effects in the DMN. I wonder if the authors could elaborate a little more on the functional role of DMN in object recognition under uncertainty.

6. Please indicate recognized vs. unrecognized outcomes in Figure 3A and its legend. It wasn't very intuitive when first looking at the figure without reading the main text.

7. It may be easier for a general reader to quickly pick up the gist of the work, if a schematic illustration of the main findings is provided, along with putative neural mechanisms.

---

## [Author Response]

Essential revisions:(1) There are some confusions in the results, such as the inconsistencies between the decoding and RSA analysis, the strange outlier pattern that might confound the results, and the ROI selection that might bias model comparison. The authors should address the points by providing convincing and thorough evidence. (See comments by Reviewer 1)

We have now included additional results showing that the decoding and RSA results are highly consistent, in the newly added Figure 2F. We have also clarified the source of the outlier patterns observed in the RDM and performed additional control analyses demonstrating that these patterns are unlikely to confound the results. Finally, we have elaborated on our rationale for the ROI selection, the choice of models, and why our ROI selection does not bias model comparison. Please see below for detailed responses.

(2) One of the key conclusions of the work, i.e., variability reduction, is not backed up by statistical evidence (reviewer 1).

Statistical analysis regarding this effect is now provided in the newly added Figure 2E.

(3) Several choices in the analysis seem arbitrary and lack verification, such as model comparison in different ROIs, and the use of two time ranges to test the hypothesis (see comments by Reviewer 1).

We now better justify the selection of models and ROIs, and we have also carried out additional analyses which show that the results of model comparisons remain qualitatively similar regardless of specific choice of time ranges (Figure 5—figure supplement 1-2).

(4) Since 7T fMRI results provide detailed spatial information as well as early recognition activities that might arise through a subcortical pathway, it seems feasible and a valuable opportunity to delineate the underlying mechanism by including subcortical regions in the analysis (see Reviewer 2's comments).

While 7T fMRI has excellent sensitivity to subcortical regions, MEG does not. Because the present study requires aligning the spatial information from 7T fMRI with the temporal information from MEG, the lack of sufficient sensitivity of MEG to subcortical signals preclude robust analysis of subcortical processing. Please see our detailed response to this point below.

Reviewer #1 (Recommendations for the authors):1. In Figure 2C, The RSA results seem to be not consistent with the decoding results shown in Figure 2B. If my understanding is correct, we would expect to see category decoding starting at 290 ms, but the RSA results are a bit uniform and do not display any clustering effect as would be the case for successful category decoding. Why is that?

The reviewer is correct that clustering effect related to the category information is not immediately apparent in the RDMs via visual inspection (Figure 2C). To address the reviewer’s concern over the potential inconsistency between results shown in RDMs and category decoding, we included an additional RSA-based analysis to provide statistical support for the clustering effect in RDMs. The results are now depicted in the newly added Figure 2F, and described in detail on p.8-9 in the Results section (also see p.22 in the methods section):

“Together with our report of image category being only decodable in recognized trials but not unrecognized trials (Figure 2B), the strong clustering effect of recognized images indicates that neural activity associated with recognized images contained sharper category information compared to unrecognized images despite the overall smaller dissimilarities among each other. To validate the compatibility between category decoding and RDM results, we ran an additional RSA-based analysis to test for the presence of category information in brain responses to recognized and unrecognized images, respectively. At each time point, we compared the representational structure within the recognized images (cells in the upper-left quadrant) and unrecognized images (bottom-right quadrant) with a category model which predicted smaller dissimilarities between images from the same category than between images from different categories (Figure 2F, right). This yielded a time course of Spearman’s rhos, indexing the strength of category information available in brain responses to recognized and unrecognized images, respectively. As expected, the result was qualitatively similar to category decoding: Significant category information occurred shortly before ~500 ms and decayed at ~1000 ms in recognized trials, whereas no category information was evident in unrecognized trials (Figure 2F, left).”

We note that minor differences between the decoding and RSA-based results, such as the transient peaks before 500 ms, are expected because they were performed at different levels of the data. The decoding analysis was tailored at the image category level, while the RSA results were based on the betweenexemplar differences. Yet, the overall similarity between decoding and RSA-based analyses on category information corroborates our interpretation that brain activity contains sharper category information when images were recognized despite the reduced cortical activity space.

2. Moreover, what do the horizontal and vertical yellow lines indicate? Do they represent some specific image exemplars that elicit abnormal MEG responses? I am concerned that they indeed contribute to the weak category decoding for unrecognized trials given that they are within the unrecognized quadrate.

The horizontal and vertical yellow lines in the RDMs corresponded to an unrecognized face (#5) and house (#1) exemplar, respectively. They showed relatively high dissimilarities to all other conditions. These high dissimilarities were likely driven by the relatively low trial numbers in these two conditions. As shown in Author response image 1, the across-subject mean trial numbers for these two conditions were 2.83 (SEM: 0.52, 5^th^ bar in ‘face U’) and 2.25 (0.43, 1^st^ bar in ‘house U’), respectively.

**Author response image 1. sa2fig1:** Across-subject mean number of trials for individual image exemplars, grouped by recognition outcome and image category. Green: recognized, red: unrecognized. Error bars represent SEM. Numbers above the bars represent across-subject mean trial number and SEM for a given image category and recognition outcome.

We do not think that the low trial numbers for these two individual conditions would have significantly contributed to the weak category decoding in unrecognized trials: In the category decoding analysis, samples for a given label (e.g., ‘face’) were pooled from trials corresponding to all five exemplars of the same category. That is, the sample size of a given label equaled the sum of trial numbers of all exemplars belonging to the that particular category. Thus, low trial number of a single exemplar had minimal effects on the overall sample size of that label which was relevant to the decoding performance. Indeed, as shown in Author response image 1, the sample sizes of both unrecognized face (mean:33.92, SEM: 2.98) and house (mean: 35.90, SEM: 2.98) lied well within the range of other recognized or unrecognized categories.

Moreover, we took additional measures to minimize potential bias due to unbalanced data. This included a resampling approach for the decoding analysis wherein label predictions were yielded from ensembles of classification models always fitted with balanced training data set, and the usage of balanced decoding accuracy to account for the unbalanced test data set (see *Method: Multivariate pattern analysis of MEG data* for details).

3. In Figure 2D, the shrinking neural representation space of unrecognizable objects does not have statistical support. The authors should do a quantitative analysis to back up the claim.

We thank the reviewer for pointing out this issue to our attention. We have now included additional results (Figure 2E) that provide statistical support for our claim of a shrinking neural representational space. These results on described on p.8 of the Results section:

“To test whether the clustering effect within the recognized images was indeed stronger than unrecognized images, we compared the mean dissimilarity between recognized images against that between unrecognized images at every time point (one-sided Wilcoxon sign rank tests). As shown in Figure 2E, the mean dissimilarity between recognized images (green) was significantly lower than between unrecognized images (red) at most of the poststimulus time points (130-1900 ms, blue horizontal bar), confirming the prominent clustering effect of recognized images over time.”

4. The authors chose ROIs based on the difference between recognized and unrecognized trials for further analysis, i.e., testing the recognition model vs. the two-state model. What is the rationale behind the pre-selection? Would such ROI selection criteria be biased toward the recognition model?

We thank the reviewer for this comment which revealed an inaccurate description about ROI selection in the previous version of the manuscript. We have now clarified that we only used the contrast between recognized and unrecognized trials to select ROIs in the higher-associative cortices (TPN and DMN regions). The basic rationale was to identify all brain regions engaged in the task, while excluding those not involved in the task. We do not think that this approach was biased towards the recognition model as the contrast itself was bidirectional and did not contain information about the expected representational structure. This was evidenced by the observation that brain activity in both the TPN and DMN regions were well explained by the recognition model despite having activation changes in the opposite directions (TPN regions showed higher *activation* in recognized trials while DMN regions showed higher *deactivation* in recognized trials). As such, in the early time window as shown in Figure 5, DAN and FPCN have higher activity in recognized trials, while DMN has higher activity in unrecognized trials, yet all three networks were better explained by the Recognition model. In addition, the best performing model for DMN flips from the Recognition model to the Two-state model from early to late time window, despite this network having a stronger deactivation for recognized trials throughout these time periods.

ROIs in the early visual cortices (EVC including bilateral V1, V2, V3, and V4) and object-selective ventral temporal cortices (VTC with face-, house-, object-, and animal-selective regions) were defined using separate retinotopic and functional localizers for each subject, independent of the main task, ruling out any potential bias towards a particular model. We have now revised p.9-10 of the manuscript regarding the ROI selection:

“Four ROIs in the early visual cortices (bilateral V1, V2, V3, and V4) and four bilateral regions in the ventral temporal cortices (VTC) selective to faces, houses, animals, and man-made objects were defined using independent retinotopy and object category functional localizers, respectively. ROIs in the frontoparietal cortices were defined based on differences in activation magnitudes between recognized and unrecognized trials (Levinson et al., 2021) and covered nine unilateral regions in the task-positive network (TPN); and eight unilateral or midline regions in the default mode network (DMN). As shown in Levinson et al. (2021), TPN regions had higher activation in recognized trials while DMN regions had stronger deactivation in recognized trials.”

5. The idea of using a "recognition model" and "two-state model" to test the representational neural format in different ROIs is interesting but unfortunately somewhat unconvincing. For example, the only difference between the two models seems to reside in the 4th quadrant, that is, the category RSA for unrecognized trials. In other words, I don't think the two models are good comparisons. Related to this point, Figure 4A is confusing since the two models work for all ROIs.

The reviewer is correct in that the recognition and two-state models differ only in the 4^th^ quadrant. However, we think this difference is important as these models represent two fundamentally different hypotheses on the neural mechanisms underlying object recognition under uncertainty: The recognition model hypothesizes that object recognition under uncertainty is facilitated via the reduction in the variability of neural activity patterns, while the two-state model hypothesizes that it is achieved via a switch between two different brain states (now better summarized in the conceptual figure suggested by Reviewer #2, Figure 6).

While it is true that the effects of both recognition and two-state model were evident across all ROIs at particular time points, please also note that our results shown in Figure 4, Figure 4—figure supplement 1-3, Figure 5, and Figure5—figure supplement 1-3 demonstrate how these two models systematically differed in the timing, strength, and sustainability of their effects across different brain regions and networks. It is on these differences that our conclusions rest.

6. Related to the above point, if the major goal is to characterize the temporal evolvement of representational models in different regions, the authors should compare the two models directly at each time point and in each ROI. This seems to be a more straightforward way.

Our choice of comparing the models at the large-scale network level was based on the observation that ROIs within each network tended to have similar model-related temporal dynamics. Accordingly, we reasoned that model comparisons at the network level would provide a concise way to summarize the results without obscuring important information. Nevertheless, we agree with the reviewer that it is also important to examine model performances at the ROI level. For this reason, we also conducted analogous model comparisons for each individual ROI (Figure 5—figure supplement 3, previously Figure S8). The results at the ROI level were largely consistent with those at the network level. In addition, we also presented the quantitative model performance time courses for each individual ROI in Figure 4B and Figure 4—figure supplements 1-3, which allows a qualitative assessment of relative model performance at a very finegrained level (each ROI, each time point). However, we believe that doing statistical model comparison at the individual ROI-individual time point level would be highly noisy and would not result in much additional information than that already presented.

7. The authors selected 0-1000ms and 1000-2000ms as the perceptual and post-perceptual time range for model comparison. This division is quite arbitrary and lacks independent criteria. Moreover, 1000 ms is too long for perceptual analysis and would certainly involve post-perceptual and feedback signals, in contrast to the perceptual claim.

To address this concern, we repeated the analysis with different choices of time ranges for model comparisons. These results show qualitatively similar results and are presented in Figure 5—figure supplement 1-2 and described on p.14 of the Results section:

“Furthermore, we repeated the same analysis using different choices of time ranges (with 100 ms and 200 ms sliding windows). As shown in Figure 5—figure supplement 1-2, the results remained qualitatively similar, providing additional empirical support for the robustness of our results.”

We reasoned that neural activity during the first 1000 ms in the post-stimulus phase was mainly involved in the perceptual process based on multiple behavioral studies using similar degraded stimuli showing long reaction times of around or above 1000 ms (for specific references, see Discussion section, p.18, last sentence in the second paragraph). In addition, work by ourselves and others has shown that feedback signals are critical for perceptual resolution of degraded images such as those used herein (low-contrast, threshold-level images), and in such contexts can take well over 500 ms to unfold (see, e.g., Flounders et al., 2019). Nonetheless, we cannot completely exclude post-perceptual signals in the first 1000 ms after stimulus onset; therefore, we have tuned down the claim on p.13 of the manuscript:

“To this end, we split the explanatory power estimates into an early (0-1000 ms following stimulus onset) and a late (1000-2000 ms) time window, reasoning that perceptual processing had relatively strong influences on brain responses in the early stage whereas those in the late stage were mainly involved in post-perceptual processing.”

8. The major findings based on commonality analysis rely on decoding results of both MEG and fMRI. Figure 2B shows that unrecognizable trials in MEG signals do not contain category information. But how about fMRI decoding results? Do the ventral visual pathway regions show successful category decoding for unrecognizable trials? If yes, could the two-state results for VTC be simply due to fMRI results?

fMRI decoding results for ROIs in the ventral visual pathway were consistent with the MEG decoding results: Category information could be decoded from BOLD responses in the EVC (V1 – V4) and VTC (category-selective areas) when participants reported the stimuli as recognizable, but not when stimuli were unrecognizable. These results were previously reported Levinson et al. (2021, Figure 4b, Figure 5c and Figure S3b therein).

Reviewer #2 (Recommendations for the authors):Wu and colleagues applied a novel MEG-fMRI fusion approach and studied the spatiotemporally resolved neural activity of object recognition under uncertainty through a reanalysis of MEG and fMRI datasets from the same research group (Podvalny et al., 2019; Levinson et al., 2021). The authors found that recognition-related brain activity occurs prior to category-related activity, and demonstrated differential representation formats (i.e., state-switch vs. variability-reduction) in ventral visual and frontoparietal regions. It was concluded that object recognition under uncertainty engages bidirectional coordination among multiple cortical networks with diverse functional roles at an early processing stage. The manuscript was very well-written. I have a few points to share as follows.1. Perhaps many readers are not familiar with the MEG-fMRI fusion approach. A few sentences may be needed to unpack the rationale and basic assumptions of this approach. For example, measurements of neural activity by different imaging modalities are presumed to reflect the same generators when they correlate on a trial-by-trial or condition-by-condition basis.

We thank the reviewer for this suggestion and have included additional text on p.9:

“We then used RSA to combine time-varying MEG data with spatially-localized fMRI data based on the correspondence between their representational structures, under the assumption that a correspondence between neural measurements by different recording modalities reflects the same neural generators (Kriegeskorte et al., 2008; Cichy and Oliva, 2020)”

2. One intriguing finding is that the onset of recognition-related activity (110-120ms according to both models) occurs much earlier than the onset of category-related activity. I wonder if the authors could discuss more about the early-stage activity. Does it suggest that the participants were aware of the content of the image at this stage? Or the neural process of early-stage activity remains under an unconscious processing stage, and the actual recognition experience occurred later at ~500ms (according to the peak decoding of recognition outcome)? A possible interpretation might be that the recognition-related activity at 110-120ms is an unconscious process associated with an initial assessment of the quality of sensory evidence, which could be accumulated over time (e.g., through corticocortical feedback activity) until the emergence of recognition experience.

The interpretation about the early-stage activity brought up by the reviewer is highly intriguing. We have now added it to the Discussion section on p.18:

“Alternatively, they may reflect the accumulation of sensory evidence preceding the rise of a particular perceptual experience (Gold and Shadlen, 2007; Kelly and O’Connell, 2015). These interpretations are not mutually exclusive and consistent with our empirical finding that the onset of recognition-related activity according to both models (significance at 110-120 ms) is earlier than the onset of category-level information (initial peak at 290 ms, significance at 470 ms).”

3. The authors mentioned that "the phase-scrambled images were not included in the analyses reported below due to comparatively low trial numbers". However, the scrambled conditions were analyzed by both Podvalny et al. 2019 and Levinson et al. 2021. Also considering that the MEG-fMRI fusion approach used in the current study was based on a condition-by-condition matrix instead of a trial-by-trial matrix, I wonder if false alarm vs. correct rejection (derived from scrambled conditions) would be information that is complementary to the current findings.

We agree with the reviewer that it would be intriguing to include scrambled image trials in the MEG-fMRI fusion. However, in contrast to real images wherein both hit and miss consisted of 20 conditions, respectively (4 categories x 5 exemplars), false alarm and correct rejection would only consist of 4 conditions (1 exemplar per category). Since the power of RSA-based analysis directly depends on the richness of condition set, the relatively small condition numbers for false alarm and correct rejection would make the comparisons with hit and miss derived from the real images or with each other difficult to interpret.

4. The early onset of frontoparietal recognition-related signals is of particular interest. The authors interpreted that they are "triggered by the fast magnocellular inputs via the dorsal visual stream or projected from the pulvinar or mediodorsal nucleus of the thalamus." As the current study used 7T-fMRI data with high spatial resolution, studying subcortical regions should be feasible. In addition, Levinson et al. 2021 identified differences in activation magnitudes between recognized and unrecognized trials in a few subcortical regions including the thalamus and brainstem. It might be worth including these subcortical regions in the ROIs.

In fact, our initial MEG-fMRI fusion analysis included subcortical ROIs reported in Levinson et al. (2021) such as the brainstem and basal ganglia-thalamus. However, the RDMs derived from these ROIs did not show positive correlations with the MEG RDMs (see Author response image 2). This is likely because MEG has poor sensitivity to subcortical sources. As such, subcortical regions were excluded from model-driven MEG-fMRI fusion analyses. The negative correlations observed between fMRI RDMs from subcortical regions and MEG RDMs (Author response image 2) are similar to that seen in V4 (Figure 3—figure supplement 1), and V4 was also excluded from our analyses for this reason.

In sum, because the present study requires aligning the spatial information from 7T fMRI with the temporal information from MEG, the lack of MEG sensitivity to subcortical signals preclude robust analysis of subcortical processing. Therefore, we excluded these regions from the analyses reported in the manuscript, as we cannot be confident about any result there.

**Author response image 2. sa2fig2:** Correlations between time-varying MEG RDMs and fMRI RDMs derived from two subcortical ROIs. Format is the same as Figure 3—figure supplements 1-3.

5. It was found that successful recognition was associated with enhanced DMN engagement despite its deactivation. There were dynamic shifts from recognition model-dominated effects to two-state model-dominated effects in the DMN. I wonder if the authors could elaborate a little more on the functional role of DMN in object recognition under uncertainty.

We have now expanded our discussion regarding DMNs’ role in our task on p.18 of the Discussion section:

“The shift towards the two-state representational structure in the late stage may indicate an enhanced intercommunication with other areas that were also dominated by the two-state model effects, such as the ventral visual regions. Speculatively, given DMN’s prominent role in both associative processing (Bar et al., 2007; Stawarczyk et al., 2021) and prior-guided perception (Gonzalez-Garcia et al., 2018; Flounders et al., 2019), this effect might reflect interactions between the current visual inputs and the observer’s prior knowledge that guide subsequent perceptual decisions and related associations.”

6. Please indicate recognized vs. unrecognized outcomes in Figure 3A and its legend. It wasn't very intuitive when first looking at the figure without reading the main text.

Figure 3A has now been modified.

7. It may be easier for a general reader to quickly pick up the gist of the work, if a schematic illustration of the main findings is provided, along with putative neural mechanisms.

We thank the reviewer for this wonderful suggestion. A schematic illustration of our main findings is now added to the Discussion section (Figure 6 on p.16).